# Calretinin positive neurons form an excitatory amplifier network in the spinal cord dorsal horn

Kelly M Smith[1,2,3,4], Tyler J Browne[1,2], Olivia C Davis[5], A Coyle[5], Kieran A Boyle[5], Masahiko Watanabe[6], Sally A Dickinson[1,2], Jacqueline A Iredale[1,2], Mark A Gradwell[1,2], Phillip Jobling[1,2], Robert J Callister[1,2], Christopher V Dayas[1,2†*], David I Hughes[5†*], Brett A Graham[1,2†*]

[1]School of Biomedical Sciences & Pharmacy, Faculty of Health, University of Newcastle, Callaghan, Australia; [2]Hunter Medical Research Institute (HMRI), New Lambton Heights, Australia; [3]Pittsburgh Center for Pain Research, University of Pittsburgh, Pittsburgh, United States; [4]Department of Neurobiology, University of Pittsburgh, Pittsburgh, United States; [5]Institute of Neuroscience Psychology, College of Medical, Veterinary & Life Sciences, University of Glasgow, Glasgow, United Kingdom; [6]Department of Anatomy, Hokkaido University School of Medicine, Sapporo, Japan

**\*For correspondence:**
christopher.dayas@newcastle.edu.au (CVD);
David.I.Hughes@glasgow.ac.uk (DIH);
brett.graham@newcastle.edu.au (BAG)

†These authors contributed equally to this work

**Abstract** Nociceptive information is relayed through the spinal cord dorsal horn, a critical area in sensory processing. The neuronal circuits in this region that underpin sensory perception must be clarified to better understand how dysfunction can lead to pathological pain. This study used an optogenetic approach to selectively activate spinal interneurons that express the calcium-binding protein calretinin (CR). We show that these interneurons form an interconnected network that can initiate and sustain enhanced excitatory signaling, and directly relay signals to lamina I projection neurons. Photoactivation of CR interneurons in vivo resulted in a significant nocifensive behavior that was morphine sensitive, caused a conditioned place aversion, and was enhanced by spared nerve injury. Furthermore, halorhodopsin-mediated inhibition of these interneurons elevated sensory thresholds. Our results suggest that dorsal horn circuits that involve excitatory CR neurons are important for the generation and amplification of pain and identify these interneurons as a future analgesic target.

## Introduction

Sensory information from the body, including nociception, itch, light touch, and thermal modalities, is first transmitted into the spinal cord dorsal horn, where this afferent input can be modulated, gated and prioritized before being relayed to higher centers for sensory perception (*Todd, 2010*; *Peirs and Seal, 2016*). It is well established that alterations to neuronal circuits within the dorsal horn can directly contribute to neuropathic and inflammatory pain, as well as persistent itch (*Basbaum et al., 2009*; *Braz et al., 2014*). Despite this being a region of immense biological importance, our understanding of the neuronal circuits associated with particular sensory modalities remains limited (*Todd, 2010*; *Peirs and Seal, 2016*). To address this knowledge gap, several groups have recently implicated neurochemically-distinct subpopulations of dorsal horn interneurons with the perception of both acute and chronic pain states (*Duan et al., 2014*; *Smith et al., 2015*; *Peirs et al., 2015*; *Petitjean et al., 2015*; *Boyle et al., 2019*).

Historically, much of the research effort on spinal circuits has focused on inhibition (*Zeilhofer et al., 2012*), and a growing number of discrete inhibitory interneuron populations have

**eLife digest** Despite being unpleasant, pain is critical to survival because it acts as a warning for damage or impending harm. Day-to-day pain like a stubbed toe or a pricked finger is called acute pain. It alerts the body to harm but only lasts a short time and usually goes away on its own. Pain that persists more than three months after the damaged tissue has healed is known as chronic pain, and it is a serious problem that is often difficult to treat. Learning more about the causes of chronic pain is necessary to help develop more effective therapies.

Nerve pathways in the spinal cord help process pain and other sensory information from the skin and relay it to the brain. These pathways include sensory fibers that carry pain information from the body to the spinal cord, as well as cells that relay this information to the brain. But not much is known about how the nerves and cells in this region prioritize or refine sensory information before sending it to the brain.

Now, Smith et al. have used mice to show that nerve cells in the spinal cord that produce the protein calretinin can act as a pain amplifier, causing it to persist. A technique called optogenetics was used to turn on calretinin nerve cells by exposing them to light. This caused the mice to behave like they were in pain even though they had not been harmed, and the behaviour stopped when they were treated with morphine, a powerful painkiller. Further experiments showed that calretinin nerve cells form a highly interconnected network in the spinal cord.

These results show that calretinin nerve cells can 'jump-start' the pain pathway within the spinal cord, even when there is no painful stimulation of the skin. Turning on these cells even briefly causes behaviours associated with prolonged pain. By revealing that networks of calretinin nerve cells in the spinal cord act like an in-built pain amplifier, the experiments identify these cells as a potential target for new treatments for chronic pain.

now been identified as substrates for sensory gating in the spinal cord (*Duan et al., 2014*; *Petitjean et al., 2015*; *Foster et al., 2015*; *Cui et al., 2016*; *Boyle et al., 2019*). In contrast, our understanding of the role excitatory interneurons play in sensory processing is far less developed. Generally, excitatory interneurons are considered to provide polysynaptic relays linking circuits dedicated to innocuous and noxious sensory input, with inhibitory populations normally modulating the passage of information through these pathways (*Takazawa and MacDermott, 2010*; *Duan et al., 2014*; *Punnakkal et al., 2014*). Such a limited role for excitatory interneurons is surprising given this population is more than double that of inhibitory interneurons in the region (laminae I and II) (*Polgár et al., 2013*). Furthermore, detailed paired recording studies have also shown that ~85% of the synaptic connections in the DH are excitatory (*Santos et al., 2007*).

Notably, we have recently shown that most calretinin-expressing (CR) neurons in laminae I and II exhibit specific electrophysiological, morphological and neurochemical properties consistent with an excitatory phenotype and respond to noxious peripheral stimulation (*Smith et al., 2015*; *Smith et al., 2016*). In fact, chemogenetic activation of CR neurons has been shown to cause nocifensive behaviors and dorsal horn activation patterns consistent with mechanical hypersensitivity (*Peirs et al., 2015*), whereas genetic ablation of these neurons can cause a selective loss of light punctate touch sensation (*Duan et al., 2014*). Prior work has established that a specific excitatory interneuron population in the deep dorsal horn that transiently express VGLUT3 relays low threshold input to CR neurons (*Peirs et al., 2015*). In this study, we aim to identify and define the postsynaptic circuits engaged by the CR population and determine the functional significance of these neurons for sensory processing and perception.

## Results

### Optogenetic activation of spinal CR⁺ neurons

To study spinal CR neuron connectivity and function in sensory processing, CR^Cre mice (Cr-IRES-Cre) were crossed with loxP-flanked-ChR2-eYFP mice (Ai32) to generate offspring where ChR2-YFP was expressed in CR neurons (CR^Cre;Ai32). The spinal distribution of YFP expression in these mice mirrored the distribution of immunoreactivity for CR (CR-IR), forming a plexus of cell bodies, dendrites

and axons that was concentrated in lamina II (*Figure 1A*). Analysis of immunolabelling patterns confirmed YFP expression was highly localized to the CR-IR population, with 78.3 ± 4% (SD) of CR-IR neurons expressing YFP (1454 cells counted in three animals), and 71.5% (±2%) of YFP-IR neurons expressing CR (1767 cells counted in three animals). These data validate spinal interneuron expression of YFP, and by extension ChR2 in the CR$^{Cre}$;Ai32 line, but also imply that a smaller population

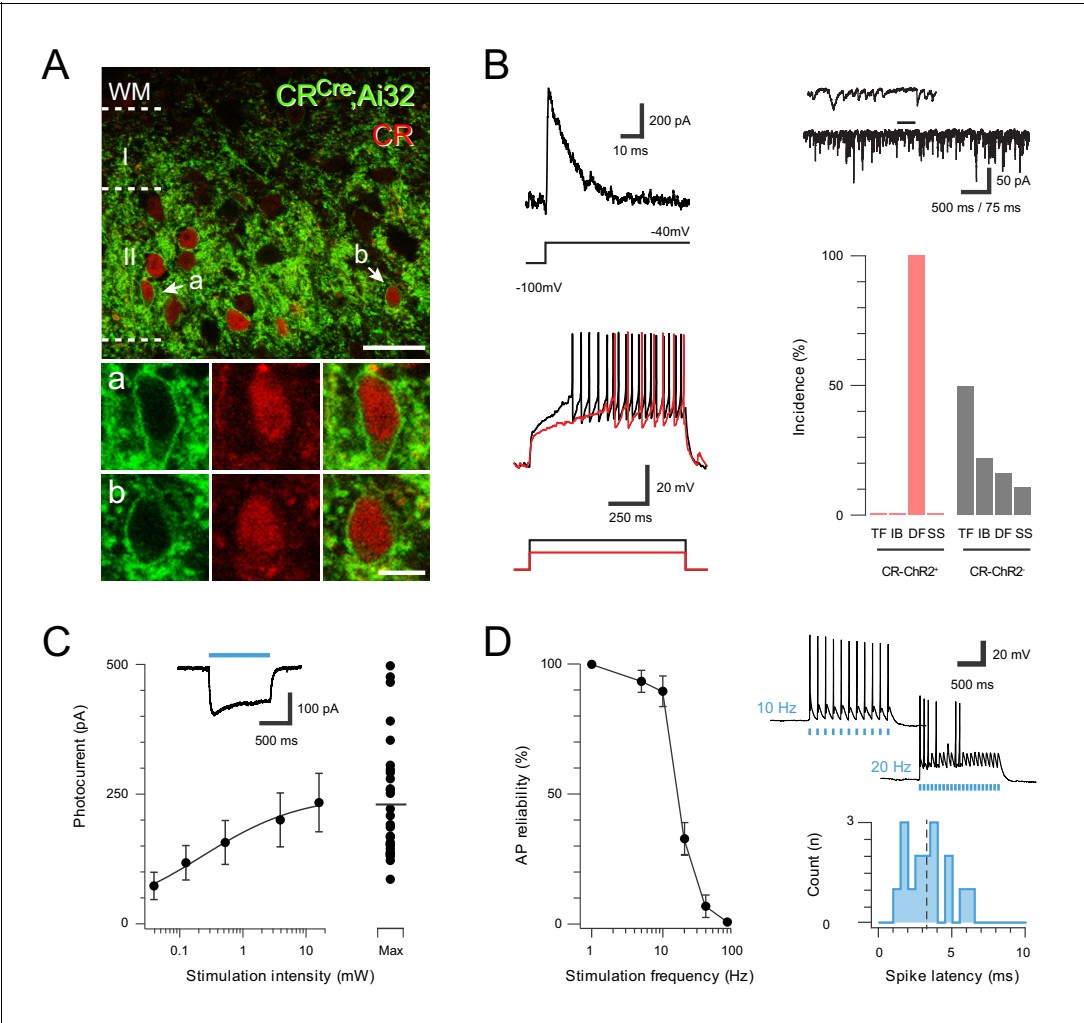

**Figure 1.** ChR2 expression in Excitatory CR$^+$ neurons. (A) Upper panel compares ChR2YFP-IR (green) and CR-IR profiles (red). There is a high degree of colocalization in LII (71 ± 2% ChR2YFP-IR neurons express CR-IR, and 78 ± 4% of CR-IR neurons express ChR2YFP-IR). Lower panels show neurons denoted 'a' and 'b' from upper panel at high magnification; ChR2YFP-IR (left), CR-IR (right), merge (center), scale = 25 μm (upper) and 5 μm (lower). (B) Excitatory CR$^+$ neurons exhibited several characteristic electrophysiological features including the voltage gated potassium current Ia (upper left, protocol below), high frequency spontaneous excitatory drive (upper right), and delayed firing (DF) discharge in response to depolarizing current injection (lower left, current step protocol below). Bar graph (lower, right) highlights the uniform incidence of DF-AP discharge in excitatory CR$^+$ positive neurons (red) when compared to a random sample of CR negative neurons (grey) in the same region (TF = tonic firing, IB = Initial bursting, DF = Delayed firing, SS = Single spiking). (C) Plot shows relationship between photostimulation intensities (0.039–16 mW) and photocurrent amplitude, error bars = SEM. Note maximum photostimulation intensity (16 mW) shows photocurrent data for individual recordings. Inset, example photocurrent response with blue bar indicating photostimulus duration. (D) Plot (left) shows reliability of evoked AP discharge at various photostimulation frequencies. APs were reliably evoked by frequencies up to 10 Hz. Representative traces (upper, right) showing reliable responses at 10 Hz but not 20 Hz photostimulation. Histogram (lower, right) shows the distribution of recruitment latency (time between the onset of photostimulation and the AP response) for CR-ChR2 recordings (dashed line shows mean of 3.2 ms).

The online version of this article includes the following figure supplement(s) for figure 1:

**Figure supplement 1.** Inhibitory CR neurons express ChR2.

**Figure supplement 2.** CR and YFP expression in sensory afferents.

of cells that express CR transiently during their development, described previously (*Peirs et al., 2015*), are also captured.

Consistent with our previous work, patch clamp recordings from YFP neurons in the CR$^{Cre}$;Ai32 line yielded characteristic electrophysiological features indicative of excitatory interneurons (*Figure 1B*). In voltage clamp, these cells exhibited robust inward photocurrents in response to photostimulation (n = 29 cells from 16 animals), which increased with stimulation intensity (0.01–16 mW, *Figure 1C*). In current clamp, photostimulation evoked AP discharge, and these neurons were able to reliably follow repetitive stimulation trains up to 10 Hz, however, reliability decreased at higher frequencies (*Figure 1D*). The latency between photostimulation onset and AP discharge (i.e. recruitment delay) across this sample was 3.29 ± 0.21 ms. We also assessed whether the subset of CR neurons identified in our previous work as inhibitory interneurons (*Atypical* CR neurons) expressed YFP (n = 13 cells from nine animals). Consistent with these earlier findings, a population of YFP expressing cells exhibited morphological and electrophysiologial features characteristic of an inhibitory phenotype (*Figure 1—figure supplement 1A*). Photostimulation in this subset of YFP expressing neurons evoked larger inward photocurrents than observed in the excitatory population (459.72 ± 34.85 pA *vs.* 233.66 ± 56.16 pA), which similarly increased with light intensity (*Figure 1—figure supplement 1B*). The inhibitory YFP population could also reliably follow repetitive photostimulation at rates up to 10 Hz, but had a shorter recruitment time than excitatory YFP neurons (2.39 ± 0.21 ms *vs.* 3.29 ± 0.38 ms, *Figure 1—figure supplement 1C*). Together, these data indicate that this CR$^{Cre}$;Ai32 mouse provides optogenetic control of both excitatory and inhibitory CR lineages (hereafter termed CR-ChR2 neurons).

Previous work has also shown that some limited expression of CR is present in the dorsal root ganglia (DRG) of rat and mouse (*Ren et al., 1993*; *Zhang et al., 2014*), suggesting this tissue should also be assessed in CR$^{Cre}$;Ai32 animals. This analysis showed GFP-labelled DRG cell bodies were occasionally observed (*Figure 1—figure supplement 2A*, left). These cells typically had large soma (mean diameter 24.5 ± 5.1 μm; n = 53 cells in 30 sections from four animals), and expressed NF200 but lacked immunolabelling for substance P. Given this finding, YFP expression was also assessed in the central terminals of several neurochemically-defined primary afferent classes in spinal cord sections. Specifically immunolabelling for VGLUT1 (myelinated low threshold mechanoreceptors; ALTMRs), substance P and CGRP (peptidergic C-fibres), prostatic acid phosphatase (Pap; non-peptidergic C-fibres), and VGLUT3 (C-fibre low threshold mechanoreceptors; CLTMRs) were assessed in tissue from CR$^{Cre}$;Ai32 animals (n = 2). Only 11 out of 815 afferent terminals counted expressed YFP-immunolabelling (*Figure 1—figure supplement 2A*, right). To support this finding, spinal cord sections from an Advillin-eGFP mouse line (Avil-EGFP) were also analysed to further determine the extent of CR-expression in the central terminals of primary afferents (*Figure 1—figure supplement 2B–F*, n = 2 animals). We found virtually no co-expression of CR-IR in YFP boutons in laminae I-III (1 out of 397), and of YFP in CR-IR terminals (2/215). In contrast, occasional examples of CR and YFP co-expression were observed in terminals located in the deep medial lamina V (*Figure 1—figure supplement 2E*), but the incidence of these profiles was not formally analysed. Together, these data rule out the expression of ChR2 in the central terminals of primary afferents arborising in laminae I-III, and support the conclusion that photostimulation of the spinal cord in our in vitro and in vivo experiments selectively recruits central CR neurons and their processes.

## CR-ChR2-activated microcircuits

Channelrhodopsin-2 assisted circuit mapping (CRACM) in the CR$^{Cre}$;Ai32 line was used to study the connectivity of CR-ChR2 neurons within dorsal horn microcircuits (*Figure 2A*). Brief full-field photostimulation (16 mW, 1 ms) was applied to assess excitatory postsynaptic responses across various dorsal horn populations (n = 73 cells from 27 animals). Strikingly, robust synaptic responses were observed in the CR-ChR2 neurons themselves (*Figure 2B*). Specifically, photostimulation of these neurons produced responses that included an immediate photocurrent and short latency optically evoked excitatory postsynaptic currents (oEPSCs) that were blocked by bath applied CNQX (10 μM). In order to analyse the oEPSCs, pharmacologically isolated photocurrents (after CNQX) were first subtracted from the original response, separating oEPSCs (*Figure 2—figure supplement 1A*). We observed oEPSCs in 96.5% of these recordings (28/29), indicating a high degree of interconnectivity in the CR-ChR2 population. A defined window for direct connection latencies was characterised by adding a delay of 2.5 ms (taken from previous paired recording studies; *Santos et al., 2007*; *Lu and*

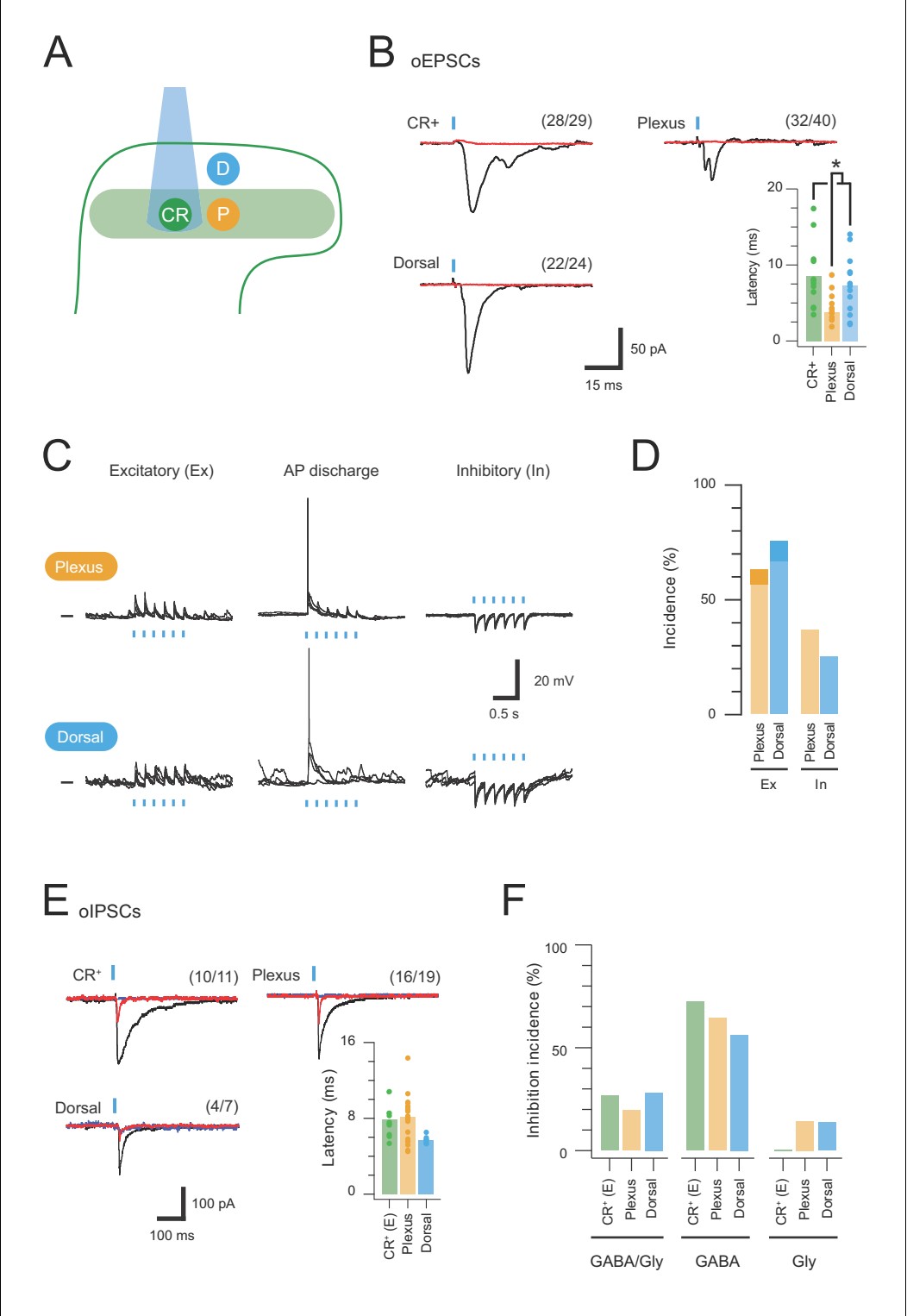

**Figure 2.** CR-ChR2 neurons provide excitatory drive throughout the DH. (**A**) Schematic shows DH populations assessed for CR-ChR2-evoked excitatory input: CR-ChR2+ neuron (green), interneurons (yellow) located within the CR+ plexus (light green shading), and interneurons located dorsal to the CR+ plexus (blue). (**B**) Photostimulation (16 mW, 1 ms) evoked robust inward currents under voltage clamp in each DH population. Traces show averaged response (black) to photostimulus (blue bar), CNQX (10 μM) abolished all responses (red). Values on each trace show number of recordings that exhibited a light induced inward current. Bar graph shows group data comparing

*Figure 2 continued on next page*

*Figure 2 continued*

oEPCS latency, which was significantly shorter in interneurons within the CR plexus (p=0.047). (**C**) Representative traces show responses during photostimulation recorded from interneurons within the CR$^+$ plexus (upper) and dorsal to this region (lower), in current clamp. In some neurons photostimulation only caused subthreshold depolarization (excitatory, left); in others depolarization evoked AP discharge (center), while in some neurons the postsynaptic response during photostimulation was inhibitory in the form of transient membrane hyperpolarisations (right). Photostimulation applied at a membrane potential of −60 mV. (**D**) Bar graphs show group data on the incidence of photostimulation responses. Darker shading denotes percentage of excitatory responses that cause AP discharge in each group. (**E**) Traces show averaged optically evoked inhibitory postsynaptic currents (oIPSCs) recorded in response to photostimulation (black trace), and following bath applied bicuculline (10 µM, red trace) and strychnine (1 µM, blue trace), left to right. Bar graph compares the latency of inhibitory responses from photostimulation onset. (**F**) Photostimulation-evoked inhibitory responses were classed as mixed (GABA/glycine, left), GABA-dominant (middle) or glycine dominant (right) based on bicuculline sensitivity. The incidence of each form of inhibition was similar across the populations assessed.

The online version of this article includes the following figure supplement(s) for figure 2:

**Figure supplement 1.** Isolation of excitatory synaptic responses in CR-ChR2$^+$ neurons and photostimulation response latencies.

**Figure supplement 2.** Isolation of inhibitory synaptic responses in CR-ChR2$^+$ neurons and photostimulation response latencies.

*Perl, 2003*) to the average AP recruitment delay for excitatory CR-ChR2 neurons (3.29 ± 0.38 ms, *Figure 1D*), allowing for AP conduction and synaptic delay. The distribution of oEPSC latencies in CR-ChR2 neurons suggested they receive both a direct and delayed input following photostimulation (35% direct, 65% delayed, *Figure 2—figure supplement 1B*).

CRACM was also applied while recording from neurons lacking ChR2-YFP, either within or dorsal to the YFP plexus (LII$_o$), showing that both plexus (32/40) and dorsal (22/24) populations received CR-ChR2 neuron input (*Figure 1B*). Using the same defined window for direct and delayed input, plexus recordings received mostly direct input (75% direct, 25% delayed, *Figure 2—figure supplement 1B*), whereas recordings dorsal to the YFP plexus exhibited a similar level of direct and delayed oEPSC input (57% direct, 43% delayed, *Figure 2—figure supplement 1B*). Comparison of oEPSC characteristics identified a significantly shorter onset of the oEPSC response for neurons within the YFP plexus compared to other populations (*Figure 2B*, plexus = 4.75 ± 0.59 ms *vs.* YFP = 8.61 ± 1.23 ms, p=0.012; dorsal = 7.34 ± 1.06 ms, p=0.047). In contrast, oEPSC time-course was similar across recordings (*Table 1*; rise time: YFP = 2.68 ± 0.41 ms; plexus = 2.89 ± 0.63 ms; and dorsal = 3.22 ± 0.64 ms. Half Width: YFP = 4.90 ± 0.62 ms; plexus = 5.40 ± 1.60 ms; and dorsal = 6.61 ± 1.16 ms). These features combined to generate similar oEPSC charge across the sampled populations (YFP = 0.66 ± 0.20 pA.s; plexus = 0.52 ± 0.17 pA.s; dorsal = 1.93 ± 0.98 pA.s).

**Table 1.** Photostimulation response characteristics.

Values are mean ± SEM. * denotes p<0.05 between cell types (CR *vs.* P *vs.* D *vs.* PN), or oIPSC type (M *vs.* G).

| Neuron type | Input | n | Latency (ms) | Amplitude (pA) | Rise time (ms) | Half-width (ms) | Charge (pA.ms) |
|---|---|---|---|---|---|---|---|
| CR-ChR2 (CR) | oEPSC | 12 | 8.6 ± 1.2 | 87.59 ± 20.04 | 2.67 ± 0.41 | 4.90 ± 0.62 | 0.66 ± 0.20 |
| Plexus (P) | oEPSC | 11 | 4.8 ± 0.6 * [CR, D, PN] | 51.42 ± 19.88 | 2.89 ± 0.63 | 5.40 ± 1.60 | 0.52 ± 0.17 |
| Dorsal (D) | oEPSC | 13 | 7.3 ± 1.1 | 126.72 ± 52.03 | 3.22 ± 0.64 | 6.61 ± 1.16 | 1.93 ± 0.98 |
| PN (PN) | oEPSC | 13 | 7.9 ± 1.4 | 70.68 ± 131.40 | 5.54 ± 1.44 * [CR, P, D] | 10.82 ± 1.60 * [CR, P, D] | 1.35 ± 0.55 |
| CR-ChR2 (CR) | Mixed-oIPSC (M) | 10 | 7.91 ± 0.53 | 146.43 ± 70.33 | 14.82 ± 3.90 | 74.39 ± 26.28 | 29.25 ± 23.87 |
| | Gly-oIPSC (G) | 10 | | 52.66 ± 12.75 | 4.36 ± 1.17 * [M] | 16.72 ± 3.42 * [M] | 9.77 ± 7.26 |
| Plexus (P) | Mixed-oIPSC (M) | 16 | 8.32 ± 0.61 | 237.16 ± 70.72 | 7.74 ± 0.60 | 71.70 ± 10.00 | 27.29 ± 9.30 |
| | Gly-oIPSC (G) | 16 | | 104.28 ± 40.55 * [M] | 6.27 ± 1.33 | 40.56 ± 19.30 * [M] | 8.17 ± 4.13 * [M] |
| Dorsal (D) | Mixed-oIPSC (M) | 4 | 5.72 ± 0.53 | 224.87 ± 76.67 | 5.99 ± 1.67 | 19.75 ± 2.93 | 17.95 ± 10.66 |
| | Gly-oIPSC (G) | 4 | | 81.34 ± 50.52 * [M] | 2.645 ± 0.35 * [M] | 15.87 ± 5.58 | 5.18 ± 3.46 * [M] |

Thus, activation of CR-ChR2 neurons produces excitation that unsurprisingly arrives first on nearby populations within the YFP plexus, before it reaches neurons dorsal to this region. In addition, interconnectivity of CR-ChR2 neurons indicates they form an excitatory network likely to enhance activity within the dorsal horn when recruited.

The impact of CR-ChR2 photostimulation on the activity of postsynaptic populations was also assessed in current clamp (*Figure 2C*, n = 22 cells, from 12 animals). Three response types were typically distinguished in these recordings; i) subthreshold excitatory responses, ii) suprathreshold excitatory responses (i.e. evoked AP discharge), and iii) inhibitory responses. Responses were assessed in neurons within the YFP plexus and dorsal to this region, but not CR-ChR2 neurons, as they were directly activated by photostimulation. The incidence of each response was similar among the YFP plexus and dorsal recordings (*Figure 2D*) including excitatory (56.3% and 66.6%) and inhibitory (37.5% and 25%) responses, with few neurons responding with AP discharge (6.2% and 8.4%).

Given the appearance of inhibitory responses in the above recordings, and the likelihood that inhibitory CR-ChR2 neurons were also activated by photostimulation, CRACM was also used to asses inhibitory connections within the dorsal horn (n = 29 cells, from 13 animals). Optically evoked inhibitory postsynaptic currents (oIPSCs) were observed in all neuron populations studied (CR-ChR2 10/11, plexus 16/19, and dorsal neurons 4/7; *Figure 2E*). The same photocurrent subtraction protocol was applied for oIPSC recordings from CR-ChR2 as above for oEPSCs (*Figure 2—figure supplement 2A*). Comparison of oIPSC characteristics showed that oIPSC latency was similar among these recordings (CR-ChR2 = 7.9 ± 0.5 ms, plexus = 8.3 ± 0.6 ms, and dorsal = 5.7 ± 0.1 ms). To determine the contribution of direct and delayed circuits to this response, a latency window for oIPSC components to be considered direct was calculated (as above for oEPSCs) using the inhibitory CR-ChR2 neuron recruitment latency of 2.39 ± 0.21 ms, and 2.5 ms to account for AP conduction and synaptic delay (*Santos et al., 2007*; *Lu and Perl, 2003*). All neuron types exhibited responses consistent with direct and delayed oIPSC components (*Figure 2—figure supplement 2B*). Delayed oIPSCs components dominated in neurons within the YFP plexus (80.5% delayed *vs.* 19.5% direct), whereas a similar mix of direct and delayed oIPSC components were recorded in excitatory CR-ChR2 neurons and neurons dorsal to the YFP plexus (excitatory CR-ChR2 = 53% delayed *vs.* 47% direct, and dorsal 59% delayed *vs.* 41% direct). Other oIPSC properties were generally similar across neuron types (*Table 1*). Importantly, as both GABA and glycine can mediate fast synaptic inhibition in the dorsal horn, sequential pharmacology was used to differentiate these neurotransmitters. Photostimulation-evoked oIPSC responses were isolated by bath application of CNQX (10 µM) and then GABAergic oIPSC components were blocked with bicuculline (10 µM), before any remaining oIPSCs were abolished with strychnine (1 µM). Comparison of oIPSCs recorded before and after bicuculline block assessed the contribution of GABA and glycine to these photostimulation responses. In this way, an oIPSC amplitude decrease of 80% or greater in bicuculline indicated GABA-dominant input, whereas a decrease of less than 20% in bicuculline indicated a glycine-dominant input. We classified oIPSCs with intermediate bicuculline-sensitivity as mixed (i.e. both GABAergic and glycinergic). Across all recordings, GABA-dominant responses were most common (*Figure 2F*: excitatory- CR-ChR2 = 72%, plexus = 65%, dorsal = 58%). Glycine dominant responses were rare, and not observed at all in excitatory CR-ChR2 neurons, with remaining cells receiving mixed inhibition. Together, these data show that in addition to a range of excitatory circuits recruited by ChR2 photostimulation, a widely distributed pattern of inhibition is also activated by CR-ChR2 neuron recruitment. Short latency direct inhibition likely comes through direct photostimulation of inhibitory CR-ChR2 neurons, whereas polysynaptic pathways recruited by photostimulation of excitatory CR-ChR2 neurons are best placed to produce longer latency indirect inhibitory responses.

## Plasticity in the CR-ChR2 network

The interconnectivity of the CR-ChR2 population and multicomponent responses to brief photostimulation (direct and delayed) suggested these neurons might be capable of producing sustained activation within dorsal horn circuits. To test this hypothesis, photostimulation duration was extended (10 s @ 10 Hz, 10 ms pulses at 16 mW) and spontaneous EPSC (sEPSC) frequency before and immediately following photostimulation were compared (*Figure 3*). Recordings in spinal slices from CR$^{Cre}$;Ai32 animals (n = 4) targeted CR-ChR2 -expressing neurons due to their coupling and predominantly excitatory phenotype, but also sampled other unidentified dorsal horn neurons, and some inhibitory CR-ChR2 neurons (differentiated from the excitatory population by their discharge

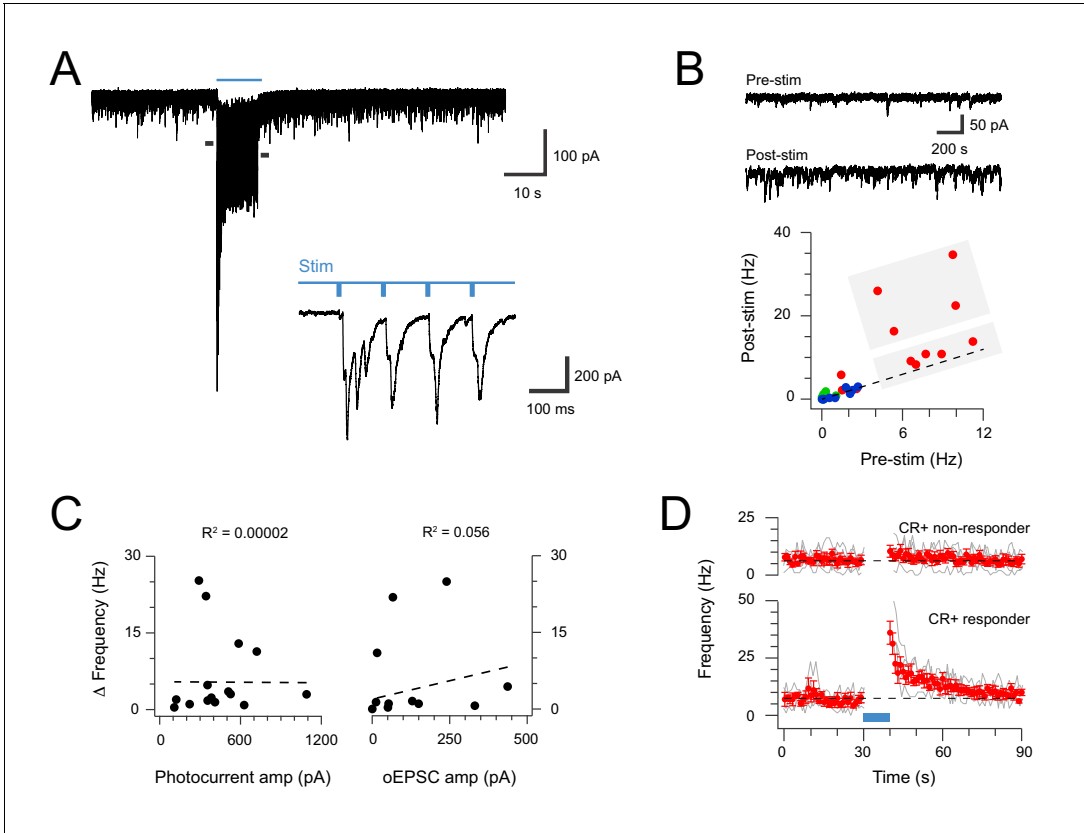

**Figure 3.** Extended CR-ChR2 photostimlation enhances spontaneous excitatory activity. (A) spinal cord slice recordings from a CR-ChR2[+] neuron showing spontaneous excitatory postsynaptic currents before, during and following full field photostimulation (blue bar, 16 mW, 10 ms pulses @ 10 Hz, 10 s). Inset shows onset of photostimulation and response on expanded time scale. Note a dramatic increase in sEPSC frequency persists following the photostimulation period. (B) Traces (upper) show 2 s pre and post photostimlation on an expanded timescale from (A). Plot (lower) shows group data comparing sEPSC frequency in the pre- and post-photostimulation (excitatory CR-ChR2 cells = red, inhibitory CR-ChR2 cells = green, unidentified DH cells = blue). Data on or near the unity line (dashed) indicates little change between pre- and post-photostimulation sEPSC frequency, however, four CR-ChR2 cells exhibited a substantial increase in post photostimulation sEPSC frequency (large grey box). (C) Plots compare pre- to post-photostimulation frequency sEPSC (Δ frequency) with photocurrent and photostimulated oEPSC amplitudes (left and right, respectively). There was no correlation between Δ frequency and either property. (D) Plots compare mean sEPSC frequency (red) across photostimulation protocol for CR-ChR2 cells deemed to exhibit a post-photostimulation increase (n = 4, post-photostimulation sEPSC frequency exceeded mean pre-photostimulation sEPSC frequency ±4 SD), and CR-ChR2 cells with a similar baseline sEPSC frequency, but no post-photostimulation change.

characteristics). These recordings exhibited a range of pre-stimulation and post-stimulation sEPSC frequency relationships, however, post-stimulation sEPSC frequency was dramatically increased in a subset of excitatory CR-ChR2 neurons (*Figure 3A–B*). A threshold of 4 standard deviations above the mean pre-stimulation sEPSC frequency was set to confidently identify recordings with increased post-photostimulation sEPSC frequency. Using this criterion one third of the excitatory CR-ChR2 cells (4/12) exhibited increased post-stimulation sEPSC frequency (*Figure 2B*). In contrast, poststimulation sEPSC frequency did not increase in unidentified dorsal horn neurons (0/9), or inhibitory CR-ChR2 neurons (0/5). While these potentiated responses could result from the specific connectivity patterns in the CR-ChR2 network, they may also relate to direct activation of photocurrents in these neurons, or the magnitude of evoked oEPSC during photostimulation. Despite this, there was no correlation between the degree of potentiation and either photocurrent amplitude (*Figure 3C* left; $r^2 = 0.00002$) or oEPSC amplitude (*Figure 2C* right; $r^2 = 0.056$). Peristimulus histograms (*Figure 3D*) compared CR-ChR2 neuron responses that exhibited increased post-stimulation sEPSC frequency (n = 4) with CR-ChR2 neurons that exhibited similar baseline sEPSC frequency but no poststimulation increase (n = 5). This highlighted the dramatic and prolonged nature of enhanced excitatory synaptic activity in the post-stimulation period, taking ~20 s before returning to baseline. Together, these

results are compatible with a model that features feedback excitation within the ChR2 network, capable of maintaining elevated excitatory signalling beyond the initial excitatory stimulus.

## Distinct dorsal horn populations are activated by CR-ChR2 neurons

To further clarify the postsynaptic targets of photostimulated CR-ChR2 neurons, we deeply anesthetized CR[Cre];Ai32 mice (n = 12) and applied direct photostimulation (10 mW, 10 ms @ 10 Hz for 10 min) to the L4/5 spinal segment(s). Animals were maintained under terminal anaesthesia, then perfused transcardially with 4% depolymerised formaldehyde in 0.1M phosphate buffer 2 hr after onset of stimulation. Spinal cord sections from photostimulated segments were subsequently processed and immunolabelled to visualise the activity marker cFos, YFP (to label CR-ChR2 cells), Pax2 as a marker of inhibitory interneurons (*Smith et al., 2015*), and three additional neurochemical markers used to differentiate neuronal populations implicated in mechanical pain pathways: Neurokinin one receptor (NK1R; *Mantyh et al., 1997*); protein kinase C gamma (PKCγ; *Peirs and Seal, 2016*), and somatostatin (SOM; *Duan et al., 2014*). Robust cFos-protein induction was restricted to the dorsal horn ipsilateral to the photostimulation probe, with the extensive mediolateral spread of cFos-IR profiles in this region (*Figure 4A–B*). Importantly, no cFos-IR nuclei were found in control animals where identical photostimulation parameters were used in CReGFP mice (n = 3 animals), confirming the specificity of photostimulation evoked Fos expression in CR[Cre];Ai32 mice. Of the cFos-IR profiles, approximately one quarter expressed YFP (28.7%; 254 out of 885 cells; n = 12 animals) and likely reflect that these were activated directly by photostimulation. The remaining cFos-IR neurons represented postsynaptic targets of the CR-ChR2 cells. Approximately 10% of these cells were NK1R-expressing lamina I neurons (*Figure 4C*. Mean proportion = 12.9%, S.E.M. ± 3.4; 16 of 124 cFos-IR cells). In these cells, immunolabelling for NK1R was confined to the cell membrane and showed no evidence of internalisation, from which we conclude that activation of these putative projection neurons results from glutamatergic synaptic input derived from photostimulated excitatory CR-ChR2 interneurons or their postsynaptic targets. In contrast, we found no evidence of cFos-expression in cells that were immunolabelled for PKCγ (0 of 477 PKCγ-IR cells), implying that ChR2-expressing cells in this mouse line do not target this specific population of excitatory interneurons (*Figure 4D*). Immunolabelling for SOM was present in ~14% of cFos-IR cell profiles (13.7% ± 2.2; 32 out of 241), however, approximately half of these also expressed YFP (50.4% ± 21.5; 15 out of 32), consistent with the expected overlap between SOM and CR in lamina II neurons (*Figure 4E*). Finally, 22% of cFos-expressing cells following spinal photostimulation were identified as inhibitory interneurons (21.3% ± 5.5; 64 out of 291), by the expression of Pax2[+] immunolabelling (*Figure 4F*), of which only ~10% of these cells expressed YFP (10.3 ± 9.2; total 4 out of 64). This confirms that inhibitory CR-ChR2 neurons are also recruited during the spinal photostimulation protocol, but that these cells are in the minority. By extension, a more substantial population of inhibitory interneurons is also engaged by activation of the excitatory CR-ChR2 population. The remaining cFos-expressing cells are most likely to be other unidentified populations of excitatory interneurons due to their absence of Pax2 labelling and may include excitatory CR[+] neurons that did not express ChR2-YFP (*Figure 4G*). The relative recruitment of each neurochemically defined population was also calculated yielding: 8.4% of all CR-ChR2 neurons (254 of 3014 YFP-IR neurons), 21% of all NK1R[+] neurons (16 of 75 neurons), 5.5% of all SOM[+] neurons (32 of 584 neurons), 0% of all PKCγ[+] neurons (0 of 477 neurons), and 15.3% of all Pax2[+] neurons (64 of 418 neurons). Taken together, these data show that activation of the CR-ChR2 network selectively recruits a diverse range of excitatory interneurons, inhibitory interneurons, and projection neurons.

## CR/SOM neurons provide direct input to projection neurons

Current models for dorsal horn microcircuitry place CR neurons in a polysynaptic circuit that signals through SOM neurons to activate lamina I projection neurons and initiate pain signalling (*Peirs et al., 2015*). Our data is compatible with this model since CR-ChR2 photostimulation evoked oEPSCs in dorsal horn neurons, including populations located superficial to the YFP plexus. This was verified further in our activity mapping data as photoactivation of CR-ChR2 cells produced robust cFos expression in CR-IR neurons, SOM-expressing interneurons, and putative NK1R-expressing projection neurons in lamina I (*Figure 4C*). Since extensive co-localisation of CR and SOM has been reported in laminae I-III previously (*Gutierrez-Mecinas et al., 2016*), it remained to be clarified how

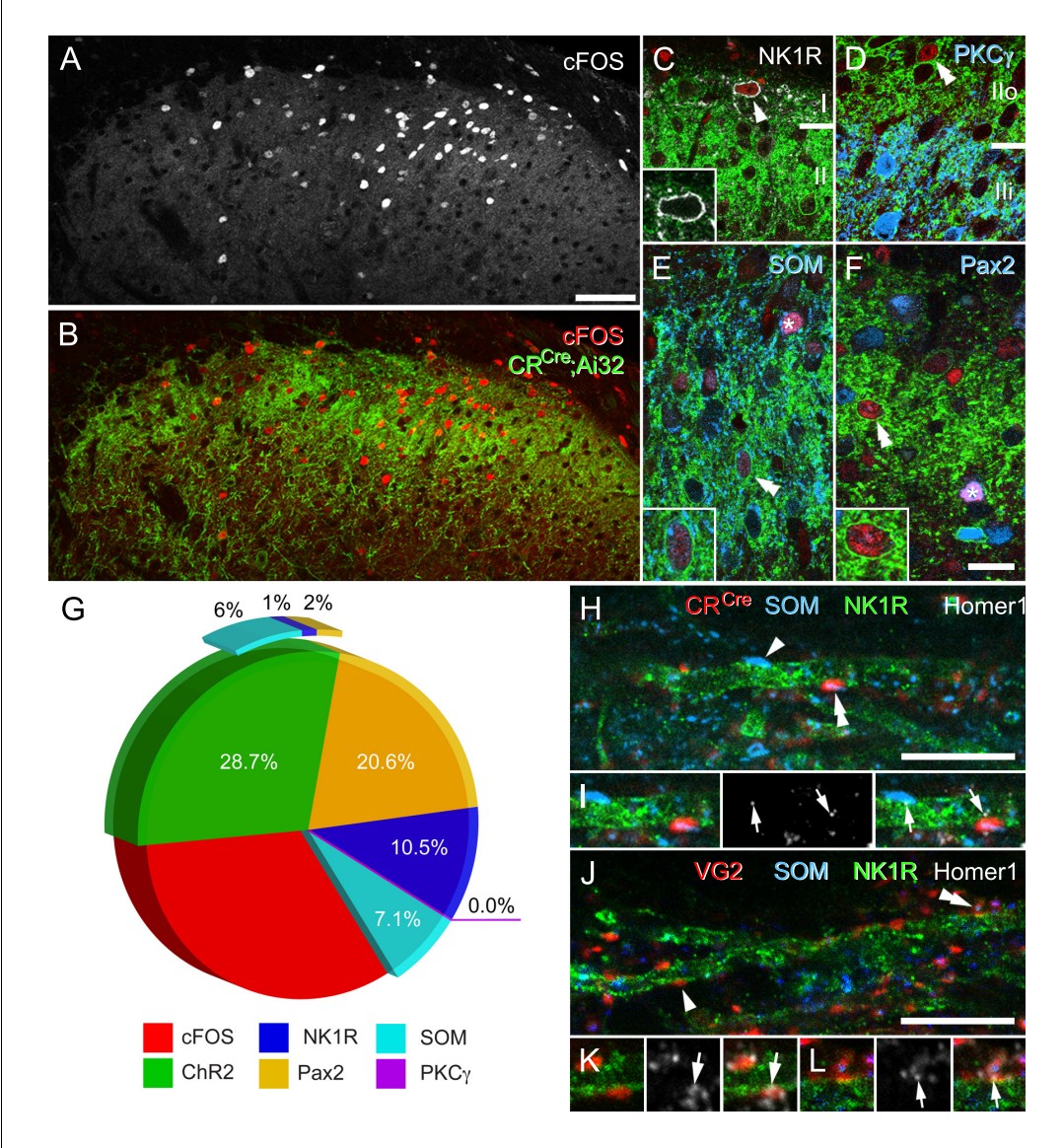

**Figure 4.** *CR-ChR2 neuron photostimulation activates multiple DH neuron populations.* (A) Following photostimulation in a deeply anaesthetized CR[Cre]; Ai32 mouse, robust cFos-IR profiles (white) were detected in laminae I and II primarily. (B) These cFOS-IR cells (red) were restricted to the ipsilateral DH, and largely confined to the CR-ChR2-YFP plexus (green). (C) Lamina I neurons often expressed cFOS, and these were commonly immunolabelled for NK1R (arrowhead; white). In these cells, NK1R-immunolabelling was confined to the cell membrane (inset). (D) Immunolabelling for cFos-IR (red) was often detected in cells that expressed YFP (green; double arrowhead), but not in cells that were immunolabelled for PKCγ (blue). (E) Many cFOS-labelled cells expressed both YFP and SOM (blue) and YFP (double arrowhead and inset), whereas others expressed only SOM (asterisk). (F) Photostimulation induced cFOS expression in Pax2-expressing interneurons (asterisk; blue), with some of these cells also showing immunolabelling for YFP (double arrowhead and inset). (G) Pie graph shows the proportion of photostimulation-evoked cFos expression accounted for by directly activating CR-ChR2 neurons (28.7%), and those populations recruited by CR-ChR2 activity including NK1R+ neurons (10.5%), SOM+ neurons (7.1%), and Pax2+ neurons (20.6%). The remaining fraction (33.1%) are likely to represent unidentified excitatory populations as they did not express Pax-2. (H–I) Most excitatory synaptic inputs on to NK1R-expressing dendrites (green) in lamina I (arrows) were derived from axon terminals immunolabelled for SOM (arrowhead; blue), many of which also originated from CR-expressing cells (double arrowhead; red). Excitatory synaptic inputs on to the dendrites of lamina I NK1R-expressing cells (green) were identified using immunolabelling for Homer 1 (white; arrows). (J–L) Most Homer puncta were directly apposed to axon terminals immunolabelled for VGLUT2 (arrowhead; I; red), many of which also co-expressed SOM (double arrowhead; J; blue). Scale bars (in μm): A, B = 100; C-F = 20; H and J = 10.

CR network activity reached projection neurons. This issue was addressed using a neuroanatomical approach with CR$^{Cre}$ mice crossed with a loxP-flanked- Synaptophysin-tdTomato reporter line (Ai34) to generate CR$^{Cre}$;Ai34 offspring where tdTomato labelled synaptic vesicles in CR-expressing cells. Tissue from these animals (n = 3 animals) was processed for immunocytochemistry to identify putative lamina I projection neurons using NK1R expression, excitatory synapses using Homer1 immunolabelling, and SOM expression to help assess CR only, SOM only, or CR/SOM inputs (*Figure 4H–I*). Using this strategy,~30% (31.5%; SD =±3.5) of all Homer puncta on NK1R dendrites were derived from CR terminals, 4.2% (±0.89) were associated with CR only inputs, and 27.3% (±3.63) were associated with boutons expressing both CR and SOM. Of all Homer puncta found on these cells, 50.4% (±0.82) were associated with SOM-IR terminals, of which 23.1% (±2.93) were SOM only, and 27.3% CR/SOM. Given that SOM immunolabelling in axon terminals is punctate and does not delineate the entire axonal boutons, we also analysed tissue from wild-type mice (n = 3 animals) in which we included immunolabelling for VGLUT2. Simultaneously visualising VGLUT2-immunolabelling in this tissue helped identify individual excitatory axon terminals, which in turn allowed us to determine the proportion of excitatory inputs on to NK1R dendrites that were derived from SOM inputs with greater precision. In this analysis, we found that 68.9% (±6.6) of all Homer puncta on NK1R-expressing dendrites in lamina I associate with VGLUT2 terminals (*Figure 4J–L*), and that most VGLUT2-IR boutons on NK1R dendrites co-expressed SOM (78.9% ± 6.28). Of all Homer puncta on NK1R dendrites, 59.4% (±8.39) were apposed by SOM-IR boutons, most of which co-expressed VGLUT2 (91.7%;±1.96). Since both SOM and VGLUT2 can be expressed in some populations of primary afferents (*Usoskin et al., 2015*), potentially contributing to the above observations we also determined the incidence of SOM and VGLUT2 co-expression in primary afferent terminals in lamina I. Lumbar spinal cord tissue sections from the Advillin-eGFP mouse (n = 2 animals) were assessed for co-expression of both SOM- and VGLUT2-immunolabelling with GFP. Only 2 of 167 GFP axon terminals in lamina I exhibited co-expression, and conversely only 4 of 304 boutons that were immunolabelled for both SOM and VGLUT2 express GFP. Together, these data confirm that CR interneurons provide substantial monosynaptic excitatory input to NK1R neurons, and that most of these terminals also express SOM. We conclude that CR and SOM interneurons represent the principal source of excitatory input to NK1R neurons in lamina I.

## CR-ChR2 neurons provide strong, direct input to Projection neurons

Given clear neuroanatomical evidence that CR neurons provided input to lamina I projection neurons (PNs) above, the functional impact of these connections was assessed. CR$^{Cre}$;Ai32 animals (n = 2) received bilateral intracranial virus injections of AAV9-CB7-Cl ~ mCherry in the parabrachial nuclei and then following a 3–4 week incubation period spinal cord slices were prepared for targeted recordings from mCherry-labelled PNs. Under these conditions, brief full-field photostimulation (16 mW, 1 ms) applied to activate CR-ChR2 neurons produced oEPSCs in PNs that were blocked by bath applied CNQX (10 μM, n = 5). oEPSCs were observed in 65% of these recordings (13/20) indicating clear connectivity between the ChR2-YFP population and PNs (*Figure 5A*). These responses could be differentiated into single oEPSC events (8/13) and multiple oEPSC responses (5/13). Consistent with some PNs receiving convergent input from several CR-ChR2 neurons, multiple oEPSC responses exhibited slower rise times and half widths than single oEPSCs (rise = 9.66 ± 2.95 ms *vs.* 2.97 ± 0.45 ms, p=0.015; halfwidth = 15.73 ± 2.16 ms *vs.* 7.76 ± 1.39 ms, p=0.016). In contrast, both oEPSC response types occurred at similar post-photostimulation latencies (8.21 ± 2.2 ms *vs.* 7.53 ± 0.7 ms, p=0.82) and together, these short latencies were comparable to those observed in other untargeted recordings dorsal to the YFP plexus and CR-ChR2 recordings (7.95 ± 1.38 ms *vs.* 7.34 ± 1.06 ms, p=0.981; and *vs.* 8.61 ± 1.23 ms, p=0.985, respectively). The strength of CR-ChR2 to PN connections was also assessed in current clamp where photostimulation-mediated oEPSPs were capable of evoking an action potential in ~70% of PNs (5/7), with a reliability of 0.675 (ie, 67.5% chance of a suprathreshold AP response). Neuroanatomical support of direct ChR2-YFP derived input to PNs was also obtained in 3 of 5 PNs that were Neurobiotin recovered, where clear YFP puncta were identified in close apposition to Neurobiotin labelled dendrites (*Figure 5B*). Together, these results demonstrate a functionally relevant monosynaptic connection exists between CR-ChR2 neurons and PNs, and is capable of recruiting PN discharge.

In light of this connectivity, the impact of repeated CR-ChR2 neuron photostimulation was also assessed to determine if this pathway supported the enhanced signalling seen in the ChR2-YFP

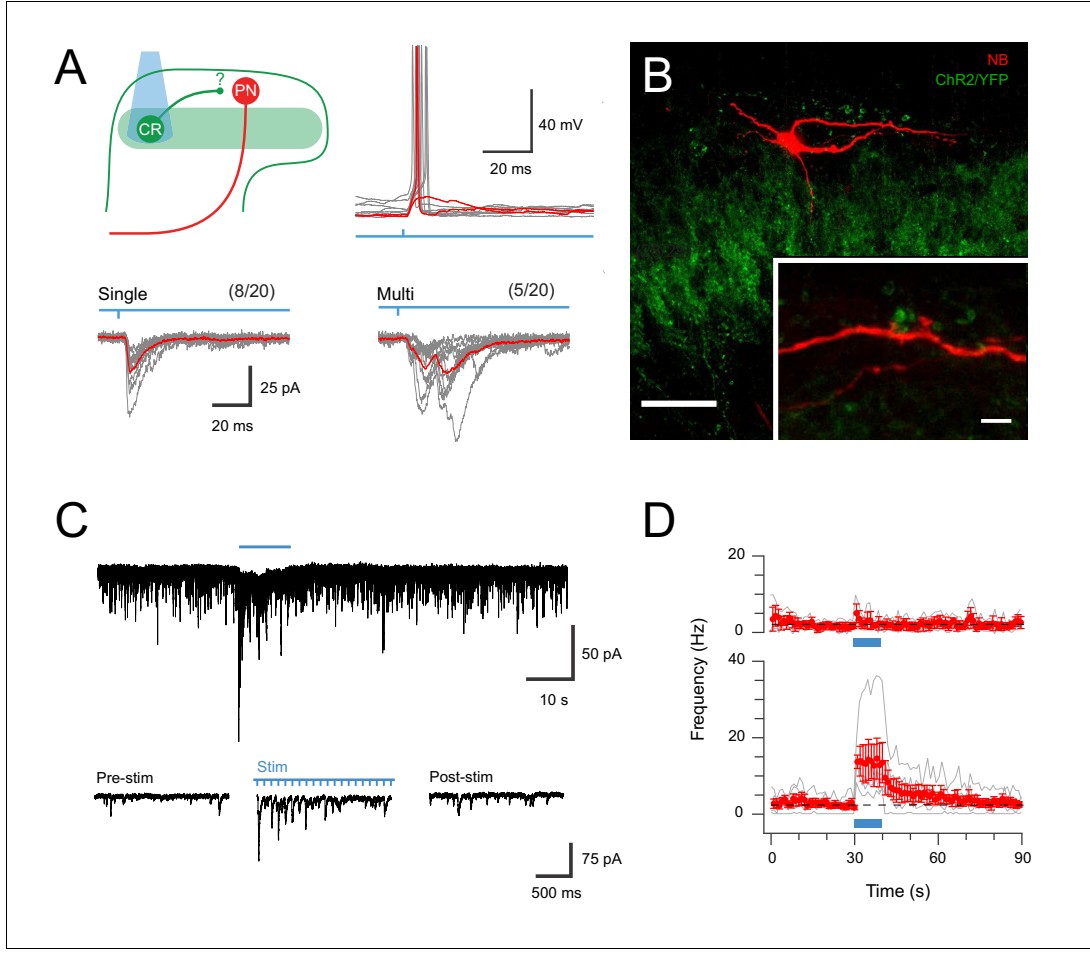

**Figure 5.** CR-ChR2 neurons provide synaptic input to LI PNs. (**A**) Schematic (upper left) shows experimental setup with CR-ChR2 neuron (CR) photostimulation (PS) applied while monitoring retrograde virus labelled projection neuron (PN) activity (n = 20 cells from two animals). Lower traces show example PS evoked inward currents recorded in PNs under voltage clamp. Responses (grey) showed either single (left) or multicomponent responses (right) during PS (blue bar) with the average response overlaid (red). Values above show number of PN recordings that exhibited PS responses. Upper right traces show an example PN recording under current clamp with PS evoked input from CR neurons able to initiate AP discharge in PNs (individual subthreshold and suprathreshold responses in red). (**B**) Image shows neurobiotin recovered PN (red) relative to expression of YFP/ChR2 in CR neurons (green). High magnification inset shows PN dendrite in close apposition with YFP/ChR2 puncta. Scale bars (in μm): 50; inset 5. (**C**) Trace shows recording from a PN with EPSCs before, during and following full field PS of CR neurons (blue bar, 16 mW, 10 ms pulses @ 10 Hz, 10 s). Insets below show EPSC activity before, during and following PS on expanded time scale. Note the increase in EPSCs during and following the PS period. (**D**) Plots compare mean EPSC frequency (red) for PNs deemed to exhibit a significant PS increase (lower, n = 5, EPSC frequency during PS exceeded mean baseline frequency ±4 SD), and PNs with a similar baseline EPSC frequency, but no PS evoked change in activity (upper, n = 3).

network (see *Figure 2*). Under these conditions ∼ 60% of PNs tested (5/8) exhibited significant and sustained responses during extended ChR2 photostimulation (10 s @ 10 Hz, 10 ms pulses at 16 mW), defined as an increase in four standard deviations above the mean background sEPSC frequency (*Figure 5C*). This increase reflected the stimulation features (i.e., approximately 10 Hz increase) and took ∼20 s to return to baseline (*Figure 5D*). In contrast, the remaining PNs still exhibited ChR2 input but did not show sustained responses (3/8). In conclusion, these results confirm that strong signalling arising from the CR-ChR2 network reaches PNs, the output cell of the dorsal horn, and therefore drive substantial output signals to higher brain regions in the ascending pain pathway.

## Photostimulation in behaving CR-ChR2 mice

The functional significance of CR-ChR2 neuron connectivity and activation within the dorsal horn was tested by chronically implanting a fibre optic probe over the surface of the dorsal spinal cord in CR[Cre];Ai32 mice for subsequent photostimulation. Following surgery, a cohort of CR[Cre];Ai32 (n = 25) animals was assessed, and shown to exhibit clear nocifensive behaviour initiated at the onset of photostimulation (10 mW, 10 ms pulses @ 10 Hz for 10 s) and outlasting the photostimulation period (*Figure 6A*). In contrast, photostimulation in a cohort of fibre optic probe implanted CReGFP (n = 9) animals did not produce photostimulation time-locked behaviours, although random grooming bouts were occasionally observed (*Figure 6A*). This data followed a series of preliminary set of experiments where behavioural responses were tested using a range of photostimulation intensities (0.5–20 mW, 10 ms pulses @ 10 Hz for 10 s; *Figure 6—figure supplement 1A–B*), which produced clear behavioural responses (*Video 1*). Specifically, responses were characterized by targeted nocifensive behaviour including paw lifting and licking/biting, typically focussed to the hind-paw or hind-limb region. The intensity and duration of these responses increased with photostimulation intensity until 10 mW and then stabilised above this (n = 5, *Figure 6—figure supplement 1A–B*). Thus, a photostimulation intensity of 10 mW was adopted for subsequent experiments, unless otherwise noted.

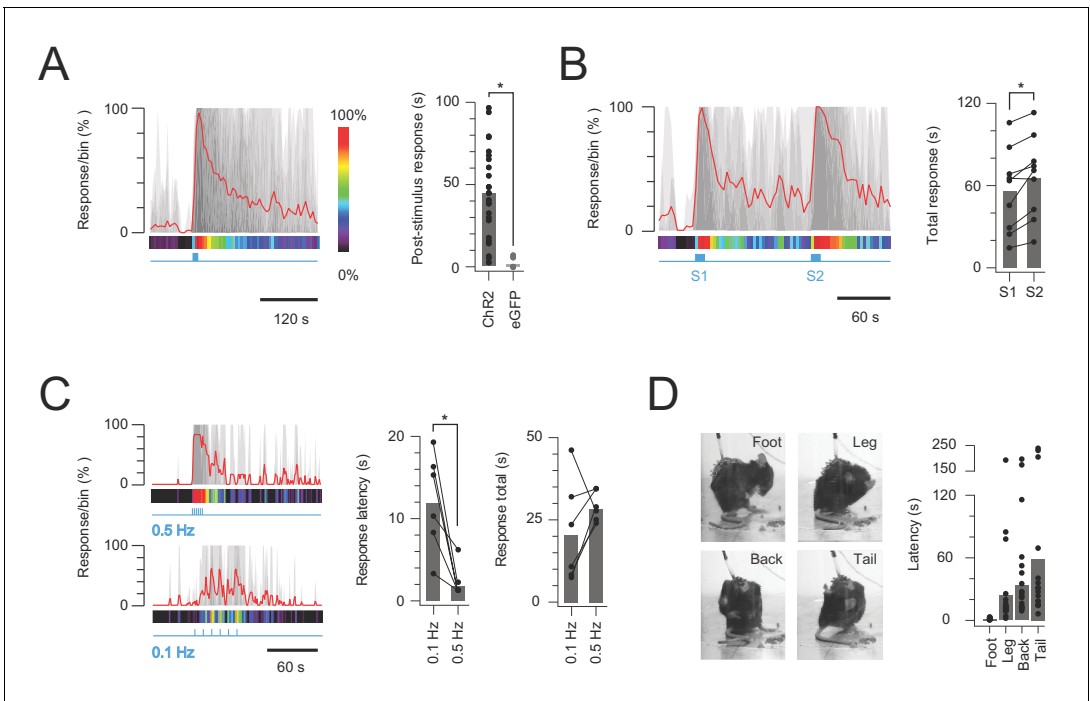

**Figure 6.** In vivo photostimulation response characteristics. (**A**) Left plot shows overlaid peristimulus histograms of photostimulation responses in CR-ChR2 mice (n = 25, grey traces) with response duration binned in 5 s epochs. Mean response is shown in red and converted to a heat bar (bins color coded to percent time groomed (red = 100%, black = 0%). Right plot compares total response duration for CR-ChR2 mice and a control group of CR-eGFP mice (n = 9). Note CR-ChR2 mouse responses varied with an average of 45 s nocifensive behavior outlasting the 10 s photostimulation period. (**B**) Left plot shows overlaid responses from a subset of animals (n = 6) that received two successive photostimuli, separated by 120 s (10 mW, 10 Hz, 10 s, grey traces). Right plot compares total response duration to initial (S1) and repeat photostimulation (S2) highlighting an average 10 s increase in the second response (p=0.005). (**C**) Left plots show overlaid responses to brief subthreshold photostimulation trains (0.72 ± 0.36 mW, 10 ms pulses) delivered at two frequencies (0.1 Hz – lower, and 0.5 Hz – upper). Right plots compare latency to photostimulation responses and total response duration at the two stimulation frequencies. Repeated subthreshold photostimuli summate to evoke a response and latency is significantly reduced by increased stimulation frequency (p=0.012). (**D**) Photostimulation responses mapped to the body region targeted in a subset of CR-ChR2 animals (paw, leg, back or tail; n = 12). Images show examples of nocifensive responses directed to the hind paw, hindlimb, back, and tail. Plots (right) compare group data for the latency of nocifensive behavior directed to different body regions. Hind paw focused responses show the shortest latency followed by significantly longer latencies for responses targeting the hind limb, back and then tail.
The online version of this article includes the following figure supplement(s) for figure 6:

**Figure supplement 1.** Photostimulation evokes an intensity dependent nociceptive behavioural response in CR[Cre];Ai32 mice.

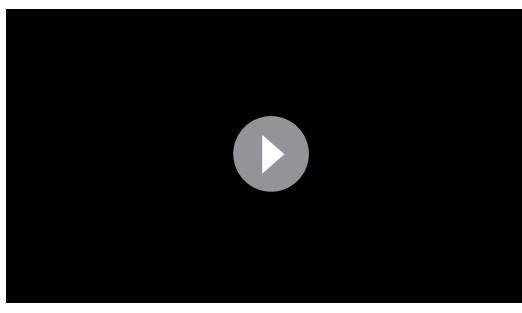

**Video 1.** Behavioural response to in vivo photostimulation of the dorsal horn calretinin network. https://elifesciences.org/articles/49190#video1

To also exclude the potential for a contribution of primary afferent signalling to behavioural responses (eg, via axon reflex), photostimulation was also assessed while the responses area (hindpaw) was under local anaesthesia (n = 4, *Figure 6—figure supplement 1C–D*). Surprisingly, photostimulation responses during local hindpaw anaesthesia (2% lidocaine) increased above baseline (control) responses (54 ± 13 s *vs.* 34 ± 4 s, p=0.05), whereas response durations during vehicle hindpaw injection were similar to controls (35 ± 10 s *vs.* 34 ± 4 s, p=0.86). This outcome is consistent with gating models of spinal sensory processing that highlight the capacity of innocuous tactile input (such as those activated by nocifensive behavioural responses during photostimulation) engaging inhibitory pathways that supress nociceptive signalling in the dorsal horn. Local anaesthesia removes this gating, allowing photostimulation evoked nociceptive signalling to continue and prolonging behavioural responses.

Given the sustained nature of post-photostimulation nocifensive responses, the potential for subsequent responses to be enhanced by prior CR-ChR2 activation was investigated by delivering two successive 10 s bouts of photostimulation, separated by 120 s (*Figure 6B*, n = 9 animals). Under these conditions the second photostimulation response was significantly longer lasting than the first (56.26 ± 10.09 s *vs.* 66.16 ± 9.95 s p=0.005; *Figure 6B*). This mirrored the observation that some CR-ChR2 neurons received sustained levels of excitatory signalling for a period following recruitment in vitro (*Figure 3*), and this signal reaches lamina I PNs (*Figure 5*). To further explore summation of the CR-ChR2 network activity, short trains of subthreshold photostimuli (which failed to elicit behavioural responses) were delivered at two frequencies (0.1 and 0.5 Hz). Photostimulation intensities that did not evoke behavioural responses during 1 s of stimulation (10 ms pulses at 10 Hz) were first established for each animal (n = 6, 0.72 ± 0.36 mW, range 0.1–2 mW). This stimulus was then repeated once every 10 s over 1 min (0.1 Hz), and once every 2 s for 12 s (0.5 Hz), altering the window for summation and associated behavioural responses without changing the total energy used to activate ChR2-YFP neurons (*Figure 6C*). These trains of stimuli reliably evoked behavioural responses at both frequencies (0.1 and 0.5 Hz) despite the absence of responses to single stimuli. The response characteristics differed, however, in that the response latency for 0.1 Hz stimulation was significantly longer than for the higher frequency (0.5 Hz) protocol (11.83 ± 2.41 ms *vs.* 2.00 ± 0.82 ms, p=0.012). The relationship between stimulation frequency and total response was more varied with 5/6 animals exhibiting longer responses for the higher frequency stimulation (0.5 Hz) but response duration falling in one animal. Thus, response duration was statistically similar in both stimulation frequencies (20.57 ± 6.31 ms *vs.* 28.73 ± 1.86 ms, p=0.218). This may be explained by the altered photostimulation frequency also changing the total stimulation duration. Regardless, these multiple stimulation paradigms reinforce the ability of CR-ChR2 neuron networks to retain subthreshold and suprathreshold excitation within an integration window that can influence the characteristics of subsequent responses.

Another pronounced feature of the CR-ChR2-evoked photostimulation response was the dynamic nature of the area targeted for nocifensive behaviour beyond the initial (primary) body region. This observation was characterized in a subset of animals (n = 18) where the initial response was directed at the right hind paw, allowing similar comparisons across multiple animals (*Figure 6D*). In this analysis, the onset of nocifensive responses directed to the right paw, right hind limb, back, and tail were measured, revealing a stereotypic progression of the nocifensive response across dermatomes. Comparison of the latency to nocifensive responses at each dermatome reinforced this stereotyped behaviour (right paw = 1.25 ± 0.22 s *vs.* right hind limb = 26.42 ± 9.11 s *vs.* back = 40.83 ± 10.78 s *vs.* tail = 59.29 ± 18.32 s, p=0.003; *Figure 6D*). Thus, in addition to sustaining excitation beyond photostimulation, optogenetic activation of the CR-ChR2 neuron network produced sensory signalling that spread to adjacent dermatomes before subsiding.

## CR-ChR2 photostimulation responses are nociceptive and aversive

Although CR-ChR2 photostimulation responses appeared nociceptive 'pain-like' in quality, additional analysis was needed to support this interpretation. First, Fos-protein activity mapping assessed the distinct distribution of cFos-expressing neurons in the brains of CR[Cre];Ai32 (n = 5) versus CReGFP (n = 5) mice following a single bout of spinal photostimulation (10 mW 10 ms pulses @ 10 Hz for 10 s). Specifically, robust cFos expression was detected in the somatosensory cortex (S1), cingulate cortex, insular cortex, and parabrachial nucleus (PBN) of CR[Cre];Ai32 mice (*Figure 7*). This expression pattern was significantly elevated above the CReGFP control group (S1 p=0.043, cingulate p=0.016, insula p=0.035, and PBN p=0.023). Neuronal activity in these regions is consistent with nociceptive signalling being relayed along the neuroaxis, mirroring the pronounced nocifensive responses observed during in vivo photostimulation.

Given nocifensive responses should also be sensitive to analgesia, a group of CR[Cre];Ai32 animals (n = 5) also underwent photostimulation during a randomized schedule of varying degrees of morphine analgesia (3 mg/kg, 10 mg/kg, or 30 mg/kg morphine s.c; or vehicle only injection of saline, s. c.). Identical photostimulation was delivered under each condition (10 mW, 10 ms pulses @ 10 Hz for 10 s) and behavioural responses analysed to provide a robust assessment of analgesic sensitivity (*Figure 8A*). Morphine produced a dose dependent reduction in the photostimulation-induced nocifensive behaviour, which was abolished at the highest morphine dose (30 mg/kg). This reinforces the nocifensive nature of the circuits activated during CR-ChR2 photostimulation.

While the above data shows CR-ChR2 photostimulation was sufficient to evoke responses consistent with nociceptive pain, this did not confirm necessity of the CR-ChR2 network in sensory-evoked responses. Thus, a photoinhibition approach was also employed by crossing CR[Cre] mice with loxP-

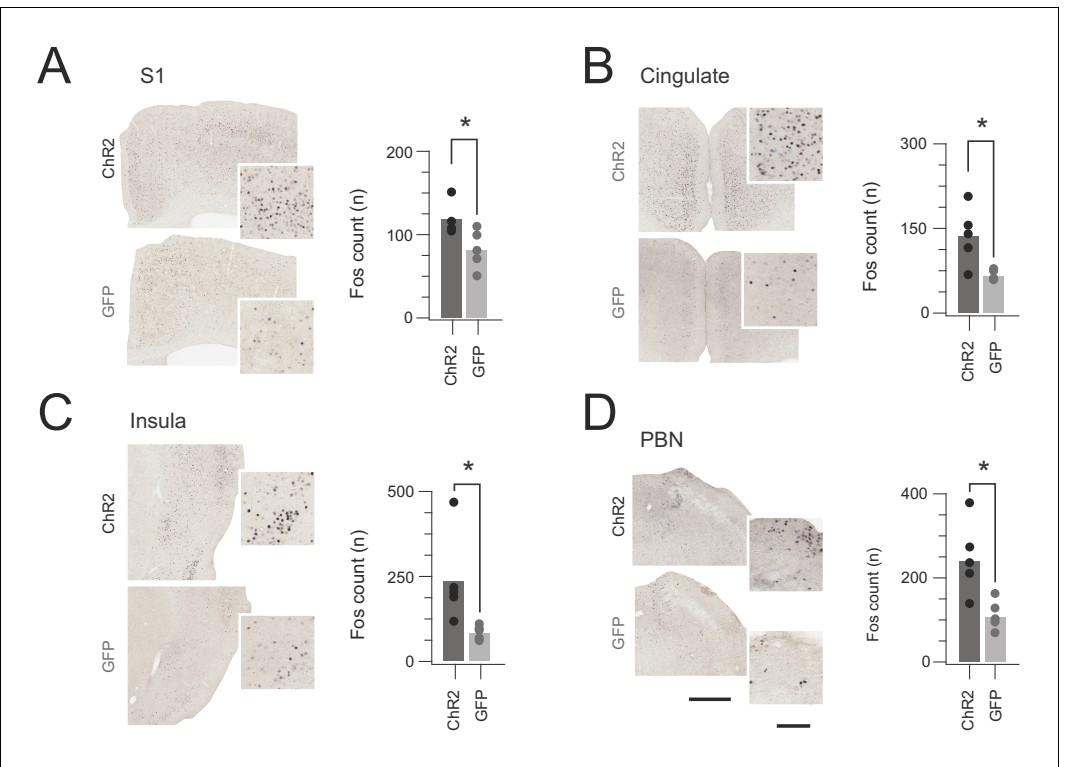

**Figure 7.** In vivo CR-ChR2 photostimulation activates nociceptive brain regions. (A-D) Images show representative brain sections from CR[cre];Ai32 (upper) and CR-GFP animals (lower) immunlabelled for Fos protein following in vivo spinal photostimulation (10 mW at 10 Hz, 10 min), insets show Fos labelled profiles at higher magnification. Bar graphs (right) of group data compare total number of Fos[+] neurons counted in corresponding brain regions. The number of Fos[+] profiles was elevated in CR[Cre];Ai32 photostimulated mice in the somatosensory cortex (S1, p=0.043), anterior cingulate cortex (Cingulate, p=0.016), Insula cortex (Insula, p=0.0346), and parabrachial nucleus (PBN, p=0.023). Scale bars = 500 μm, inset = 100 μm.

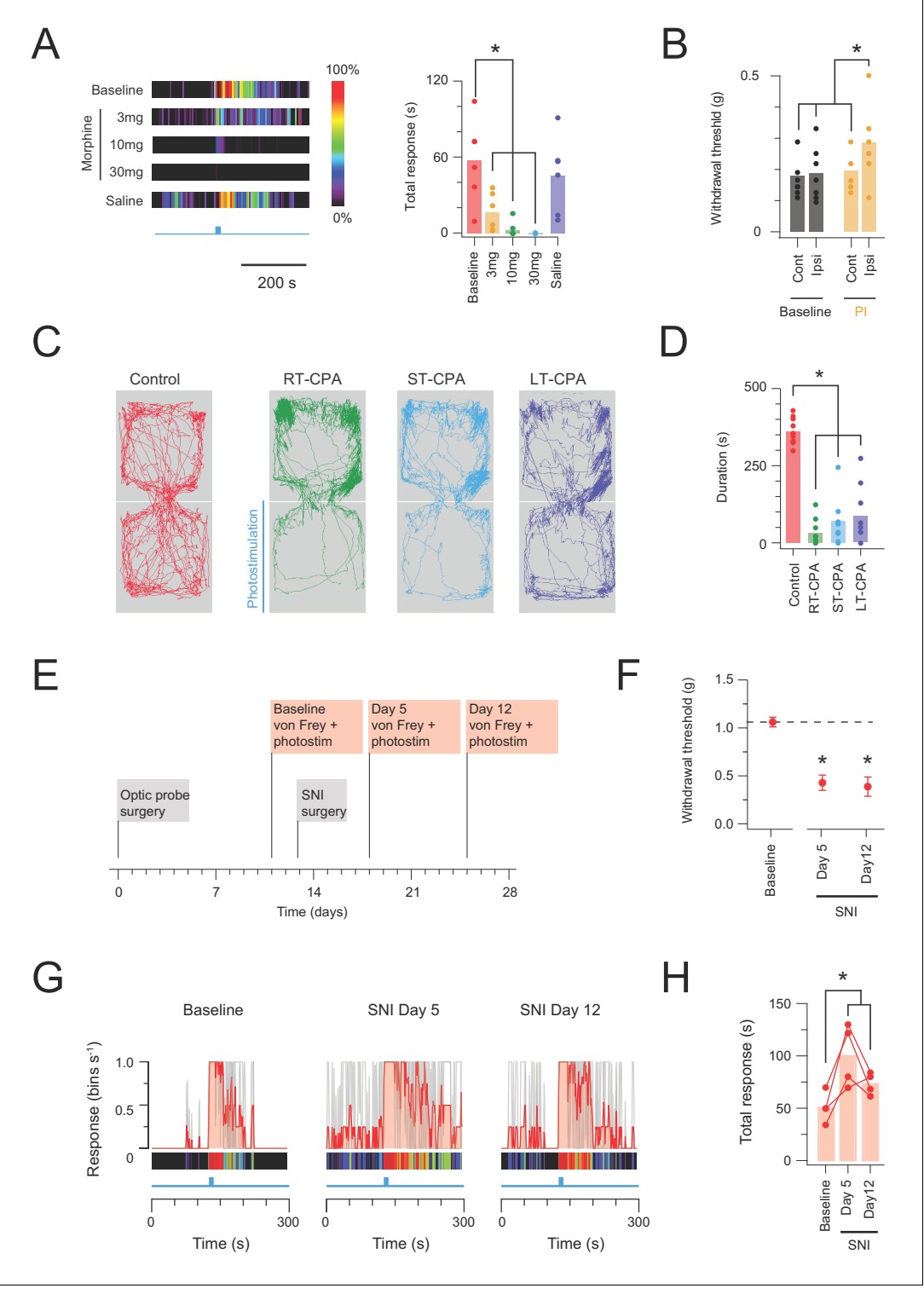

**Figure 8.** In vivo CR-ChR2 photostimulation has nociceptive qualities. (**A**) Heat bars (left) show average photostimulation responses (10 mW at 10 Hz, 10 s), from a subset of animals (n = 6) assessed under baseline conditions, after, three morphine doses (3, 10, and 30 mg/kg i.p.), and control saline injection. Bar graphs (right) compares group data for total nocifensive response duration under each condition. Morphine produced a dose dependent block of the nocifensive response compared to baseline (3 mg/kg, p=0.024; 10 mg/kg, p=0.002; 30 mg/kg, p=0.001). (**B**) Group plots compare mean withdrawal threshold (von Frey) for hind paws ipsilateral and contralateral to spinal fiber optic probe placement. Left bars show mean withdrawal threshold under baseline

*Figure 8 continued on next page*

*Figure 8 continued*

conditions and right bars show mean withdrawal threshold assessed with NpHR3-mediated photoinhibition of CR$^+$ neurons in the ipsilateral spinal cord. Spinal photoinhbition selectively increased withdrawal threshold for the ipsilateral hindpaw, consistent with CR$^+$ neuron inhibition decreasing mechanical sensitivity (cont = contralateral, ipsi = ipsilateral). (C) Four maps (left) show mouse activity traces during the conditioned place aversion (CPA) testing under: control conditions (no photostimulation - red trace); real time CPA training (RT-CPA – green trace) where one arena is assigned for photostimulation (10 mW, 10 Hz, 10 s in every minute) on entry; short-term CPA (ST-CPA), assessed 1 hr following the RT-CPA session with no photostimulation (blue trace); and long-term CPA (LT-CPA), assessed 24 hr after the RT-CPA session again with no photostiulation (purple trace). (D) Bar graph (right) compares time spent in the photostimulation arena during CPA testing. Spinal photostimulation of CR-ChR2 neurons established a robust real-time conditioned place aversion by the fourth RT-CPA session (green) significantly reducing time in the photostimulation arena (p<0.0001). These reductions persisted during ST-CPA (p=0.0001) and LT-CPA (p=0.0001). (E) Timeline shows experimental sequence for testing CR-ChR2 photosimulation responses before and after spared nerve injury (SNI) surgery. (F) Plot shows 50% withdrawal thresholds for CR$^{Cre}$;Ai32 mice at baseline and following SNI surgery. All animals developed mechanical hypersensitivity. (G) plots show overlaid peristimulus histograms of photostimulation responses in CR$^{Cre}$;Ai32 mice (n = 4, grey traces) at baseline, and 5 and 12 days following SNI surgery. Mean response is shown in red and converted to a heat bar (bins color coded to percent time groomed (red = 100%, black = 0%). (H) Plot shows individual photostimulation behavioral response durations at baseline, and 5 and 12 days following SNI surgery (bar show mean). Responses were significantly longer post-SNI.

The online version of this article includes the following figure supplement(s) for figure 8:

**Figure supplement 1.** Halorhodopsin-mediated photoinhibition of CR$^+$ neurons in CR$^{Cre}$;Ai32 mice.

flanked-NpHR3YFP mice (Ai39) to generate offspring where halorhodopsin (NpHR3YFP) was expressed in CR$^+$ neurons (CR$^{Cre}$;Ai39). To validate YFP expression in CR cells, we assessed the incidence of co-expression of these markers in spinal neurons in laminae I and II (n = 3 animals). We found that 82.2% (±1.27) of YFP-expressing cells were immunopositive for CR (967 YFP cells analysed; range 281, 316 and 370 cells per animal; *Figure 8—figure supplement 1A*), and that 94.1% (±4.27) of CR-IR cells expressed YFP (225 CR-IR cells analysed; 58, 78 and 89 cells per animal). Furthermore, full-field illumination (590 nM, 20 mW) evoked prominent outward currents consistent with a NpHR3-mediated potassium conductance (*Figure 8—figure supplement 1B*). NpHR3-mediated photoinhibition of AP discharge was confirmed by comparing NpHR3 neuron spiking responses to depolarizing current injection with and without photoinhibition (*Figure 8—figure supplement 1C*). Under these conditions photoinhibition increased the rheobase current required to activate AP spiking and decreased the number of APs evoked during increasing current steps (*Figure 8—figure supplement 1C*). Since chemogenic activation of CR neurons has been show to lower mechanical but not thermal thresholds (*Peirs et al., 2015*), we assessed the responsiveness of CR$^{Cre}$;Ai39 mice to mechanical stimulation during NpHR3-mediated photoinhibition in vivo. Mice (n = 8) were implanted with spinal fibre optic probes and underwent mechanical threshold testing using von Frey

filaments (*Figure 8B*). Photoinhibition significantly increased paw withdrawal thresholds on the ipsilateral (photoinhibited) hind paw (0.19 ± 0.02 g *vs.* 0.29 ± 0.04 g, p=0.017) but not contralateral side (0.18 ± 0.02 g *vs.* 0.19 ± 0.03 g, p=0.721), supporting a role for the CR$^+$ network in processing of noxious tactile sensation.

The relative potency and valence of spinal photostimulation was assessed in a group of CR$^{Cre}$;Ai32 animals (n = 13) with spinal fibre optic probes implanted that subsequently underwent conditioned place aversion testing (*Figure 8C–D*). Baseline preference for each animal was determined in a two-arena enclosure without photostimulation, and the preferred arena was then assigned for photostimulation

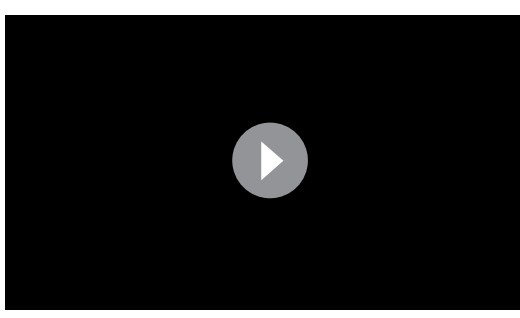

**Video 2.** In vivo photostimulation of the dorsal horn calretinin network causes a real time conditioned place aversion.

https://elifesciences.org/articles/49190#video2

(10 mW 10 ms pulses @ 10 Hz for 10 s in every min). Animals were subsequently tested for a real-time place aversion (RT-PA) over four sessions, i.e. learned avoidance of the photostimulation arena (*Video 2*). Animals that exhibited a strong RT-PA on the last two sessions, defined as a 50% reduction from baseline in time spent in photostimulation area, were subsequently tested for a traditional conditioned place aversion (CPA) 1 hr (short term) and 24 hr (long term) after the last RT-PA session (9/13 animals). Importantly, no photostimulation was delivered during this CPA testing, instead assessing the aversive nature of photostimulation recall. In short term CPA testing (ST-CPA), animals retained a significant aversion to the previous photostimulation arena compared to baseline (363 ± 15 s *vs*. 72 ± 25 s, p=0.0001). Likewise, in long term CPA testing (LT-CPA) aversion to the previous photostimulation arena was still apparent 24 hr after RT-PA (363 ± 15 s *vs*. 89 ± 31 s, p=0.0001). In summary, all CPA testing revealed a clear preference for the no-photostimulation arena, with behavioural traces also suggesting that animals spent more time stationary in this area (*Figure 8C*), potentially reflecting freezing behaviour during these tests. Thus, spinal photostimulation of the ChR2 population produced a potent sensory experience with a strong and lasting negative valence.

Finally, to assess a potential role for the CR-network in pathological pain processing, in vivo photostimulation responses were assessed in a group of CR$^{Cre}$;Ai32 animals (n = 4) before and after spared nerve injury surgery (*Figure 8E–H*). Importantly, this analysis only included animals with photostimulation responses directed to the hind paw including the spared (sural nerve) territory. As expected, SNI reduced ipsilateral hind paw mechanical withdrawal thresholds at days 5 and 12 post-surgery (SNId5 and SNId12), consistent with the development of neuropathic mechanical allodynina (*Figure 8F*). Regarding in vivo photostimulation, these animals exhibited robust nocifensive responses at baseline (pre-SNI surgery) that were comparable to previous experiments. Following SNI, however, the duration of photostimulation responses was significantly elevated (50.55 ± 10.91 s vs. 97.06 ± 20.40 s and 72.43 ± 7.08 s, baseline vs. SNId5 and SNId12; n = 4, p=0.018) implicating the CR network in enhanced dorsal horn excitability under neuropathic conditions (*Figure 8G–H*). In contrast, photostimulation responses remained stable in sham-operated CR$^{Cre}$;Ai32 animals (46.03 ± 4.71 s vs. 39.49 ± 4.08 s and 49.25 ± 19.27 s, baseline vs. SNId5 and SNId12; n = 2).

## Discussion

Our limited understanding of sensory coding in the spinal cord remains a significant barrier to defining how normal sensory experience evolves and how pathological conditions such as chronic pain develop (*Todd, 2010*; *Hachisuka et al., 2018*). In this study we applied both in vitro and in vivo optogenetic approaches and show that a specific population of dorsal horn interneurons that express CR form a highly interconnected excitatory network that is capable of driving excitation in multiple postsynaptic populations, including projection neurons via direct synaptic input. Excitation of this microcircuitry outlasts initial CR-ChR2 neuron activation, indicating that recruitment of these cells has the capacity to sustain synaptic activity within the network of reciprocal excitatory connections. Extending these in vitro findings, optogenetic activation of the CR-ChR2 population in awake animals caused a profound and multifaceted nocifensive response, indicating that this network can prolong, spread and amplify spinal nociceptive signaling. Together, these findings provide a detailed examination of the excitatory CR-ChR2 interneuron population and the postsynaptic circuits they activate during sensory processing.

### CR neuron microcircuits

Our in vitro electrophysiology showed that CR-ChR2 neurons exhibit diverse functional excitatory synaptic connections within the dorsal horn. A surprising finding in this work was the high degree of interconnectivity between CR-ChR2 cells, establishing an excitatory network that when recruited, could substantially enhance excitation in the dorsal horn (*Figures 2* and *3*). This is in line with observations using viral-mediated excitatory DREADD-expression to activate CR neurons selectively (*Peirs et al., 2015*). This work showed that CNO exposure activated a substantial proportion of transduced DREADD-positive CR neurons (30%), however, an additional large proportion of DREADD-negative CR neurons (45%) were also activated. This observation supports the capacity of CR neuron recruitment to drive activation of the wider CR network. Such interconnectivity in a neurochemically defined population has been described for another dorsal horn population identified by

expression of neurotensin (*Hachisuka et al., 2018*), although this work did not go on to demonstrate the functional relevance in behaving animals. Nevertheless, this work highlighted how interconnected excitatory networks could potentiate excitatory signaling in the dorsal horn. Likewise, similar arrangements of interconnected neuronal networks have been reported in other CNS regions, albeit typically involving inhibitory populations (*Tamás et al., 2000*; *Meyer et al., 2002*; *Woodruff and Sah, 2007*). This interconnectivity among excitatory interneurons suggests the CR-ChR2 network we have targeted here represents a potent source of excitatory signaling that could amplify afferent input in the dorsal horn.

CRACM experiments also showed that input from CR-ChR2 neurons is widespread, with other neurons located in the YFP-expressing plexus as well as more dorsal populations, receiving short and longer latency input following CR-ChR2 activation. These experiments also confirmed that lamina I projection neurons, which often received convergent, multicomponent inputs and more reliably discharged APs in response to this input than other populations, are also a target of these cells. These features are consistent with CR-ChR2 neurons initiating excitation that converges on projection neurons. Such excitatory relays have been proposed in the dorsal horn as a substrate for allodynia, where low threshold mechanical inputs are transmitted into nociceptive circuits (*Torsney and MacDermott, 2006*; *Miraucourt et al., 2007*; *Neumann et al., 2008*; *Peirs et al., 2015*; *Boyle et al., 2019*), as well as feed forward circuits that supplement excitation during nociceptive processing (*Lu and Perl, 2005*; *Lu et al., 2013*). For example, under pathological conditions CR neurons have been reported to receive low-threshold (Aβ afferent) input via a population of lamina III neurons that transiently express VGLUT3 and relay this information into nociceptive circuits (*Peirs et al., 2015*). This work also used cFos labelling and DREADD silencing of various populations to propose a circuit connecting CR neurons to projection neurons in lamina I via polysynaptic connections that included an interposed population of SOM interneurons. In support of this proposal, we find approximately 30% of all excitatory input to lamina I projection neurons are derived from axons of CR interneurons (*Figure 4*), of which a significant proportion also express SOM (86.5%;±3.17). These findings demonstrate that CR neurons provide considerable direct input to the lamina I projection neuron population, and when considered alongside the CR-ChR2 network interconnectivity, these have the potential to greatly influence spinal sensory outputs. Our findings are in agreement with previous studies which have reported that CR interneurons activate SOM-expressing interneurons, but not those that express PKCγ (*Peirs et al., 2015*), and we also demonstrate that CR-expressing interneurons can recruit lamina I projection neurons either directly, or via polysynaptic circuit incorporating SOM neurons (*Figure 4*). It is important to note that methodological approaches may account for subtle differences in the circuitry we define compared to those studies where intraspinal injections of cre-dependent viruses have been used (*Peirs et al., 2015*), however, the fidelity of the YFP expression we see in CR immunolabelled cells in our CR[Cre];Ai32 mice is nonetheless similar to those reported in spinally-injected animals (*Peirs et al., 2015*). The pattern of YFP expression, absence of YFP labelling in neurochemically-defined sensory afferents in laminae I-IV and the absence of NK1R-internalisation in lamina I cells following in vivo photo-stimulation, support the view that aberrant recruitment of primary afferents does not occur in our experiments. Rather, behavioral responses and spinal activation profiles result from the central activation of ChR2-expressing cells. In summary, the interconnectivity of CR neurons, combined with the postsynaptic circuitry we have identified provides a mechanism to relay low threshold input into nociceptive circuits (allodynia), and provide additional excitation during nociceptive processing (hyperalgesia) through reverberating patterns of excitation.

Inhibitory signaling has been also a central element in models of spinal sensory processing since publication of the gate control theory and contemporary work from a number of groups has since identified several critical inhibitory populations (*Duan et al., 2014*; *Foster et al., 2015*; *Petitjean et al., 2015*; *Cui et al., 2016*). Consistent with these views, our cFos mapping also supports a role for the CR network in engaging inhibitory interneurons. These inhibitory circuits may be important for modality coding by suppressing selective populations while the CR circuits are active (*Zeilhofer et al., 2012*; *Price and Prescott, 2015*). Such inhibition is known to act in other sensory systems to refine receptive field characteristics and a similar constraint over the activation of CR neurons would fit such a model (*Woolf and Fitzgerald, 1983*; *Kato et al., 2011*). Finally, ongoing inhibition can tune sensory thresholds, as demonstrated in several studies where diminished inhibition has

been shown to lead to pathological conditions such as chronic pain and itch (*Moore et al., 2002*; *Coull et al., 2003*; *Zeilhofer, 2005*; *Ross et al., 2010*).

An important distinction between the current study and previous work is that the inhibition evoked in our study is driven exclusively by recruitment of CR spinal interneurons, rather than as a result of primary afferent drive. Thus, the CR related microcircuits appear to have an in-built mechanism to limit the outcome of their activity under control conditions. By extension, any reduction to this inhibition would unmask added excitation with dorsal horn circuits with relevance to many pathological conditions that feature aberrant excitation. We show that CR network evoked inhibition is widespread in the dorsal horn, with a range of latencies that indicate both direct and indirect circuits are engaged (*Figure 2—figure supplement 2D*). Direct inhibition is not surprising given we have previously described a small inhibitory CR population (*Smith et al., 2015*), and show here that they express ChR2 in the CR^Cre;Ai32 animals used. These neurons are directly activated during spinal photostimulation and therefore provide the only source of short latency monosynaptic inhibition. In contrast, secondary recruitment of other inhibitory populations by the excitatory CR neurons produces longer latency polysynaptic inhibition. Previous work has shown dynophin-expressing neurons provide important gating inhibition to somatostatin neurons (*Duan et al., 2014*), and given the overlap between somatostatin and CR, dynorphin cells are therefore one candidate population for the polysynaptic inhibition observed here. Similarly, GABAergic enkephalin neurons have recently been implicated in gating mechanical pain (*François et al., 2017*) and thus may also contribute to CR network evoked inhibition.

## Behavioral consequence of CR-ChR2 activation

The behavioral consequence of experimentally activating CR circuits in vivo is striking, with a targeted and multifaceted nocifensive response (paw licking, biting, lifting, and shaking) directed to dermatomes predicted by the location of fiber optic implant and initiated upon photostimulation. This response profile is consistent with our previous work showing that CR neurons respond maximally to noxious mechanical stimulation, with some activation also resulting from hindpaw injection of capsaicin (*Smith et al., 2015*). CR neuron photostimulation responses also resemble reports from two other groups that have applied spinal photostimulation to populations of dorsal horn neurons (*Christensen et al., 2016*; *Bonin et al., 2016*). The most relevant assessed excitatory somatostatin neurons, reporting photostimulation produced abrupt nocifensive behavior and a conditioned place aversion (*Christensen et al., 2016*). This conserved behavioral profile is again consistent with the overlap in somatostatin and CR populations studied and the postsynaptic position of somatostatin neurons relative to the CR population. An additional observation in the somatostatin photostimulation study was the presence of itch related behaviors. We confirmed the nociceptive nature of CR photostimulation by demonstrating selective cFos labelling in distinct pain processing brain nuclei as well as sensitivity of the photostimulation responses to morphine (*Figures 7* and *8*). We observed a dose dependent inhibition of photostimulation related behavior that was abolished at the highest morphine dose. Morphine is a prototypical analgesic but does not have antipruritic actions, with intrathecal administration known to induce itch (*Lui and Ng, 2011*). Furthermore, we did not see scratching responses in our experiments, however, future studies employing specific itch assays would be required to further validate these observations.

It is worth noting that a small number of inhibitory CR neurons were also activated during photostimulation (*Figure 4G*). Previous optogenetic manipulation of inhibitory populations in the dorsal horn has only been reported using archaerhodopsin, which produced a predictable decrease in sensory thresholds (*Bonin et al., 2016*). The related approach of chemogenetic activation has, however, been applied to activate parvalbumin inhibitory interneurons in the dorsal horn, demonstrating an increase in sensory thresholds as well as an attenuation of nerve injury induced allodynia (*Petitjean et al., 2015*). Taken together, these results support the well-established role of inhibitory populations in suppressing spinal nociceptive signaling and suggest the recruitment of some inhibitory CR neurons in our experiments will have had minimal effects on the robust behavioral outcomes attributed to the excitatory CR population. Furthermore, we show that halorhodopsin-mediated inhibition of the CR population increased mechanical withdrawal thresholds, consistent with a result that would be predicted for inactivation of an inhibitory population from the above work.

Three behavioral observations during CR neuron photostimulation warrant further discussion. First, behavioral responses persisted well beyond the termination of photostimulation (*Figure 6A*).

This observation was paralleled by a sustained increased in spontaneous excitatory activity in CR neurons and LI projection neurons under extended in vitro photostimulation conditions, indicating this excitatory network can sustain excitatory activity following recruitment. Second, behavioral responses to repeated photostimulation were enhanced, indicating short-term plasticity within the CR network (*Figure 6B–C*). Previous work has focused largely on plasticity between primary afferents and dorsal horn populations (*Baba et al., 2001*; *Luo et al., 2014*), whereas our findings extend our understanding to show plasticity also occurs within intrinsic dorsal horn circuits. Finally, photostimulation responses showed a predictable pattern that initially saw nocifensive behavior focused on a specific body region (commonly the paw) but then progressed over a number of adjacent dermatomes (*Figure 6D*). This coupled with spinal cFos activation patterns, which extended over the mediolateral extent of the dorsal horn (*Figure 4A–B*), suggest the excitatory CR network provides a pathway to spread excitation across normal dermatome boundaries. Similarly, though not examined here, photostimulation responses imply that the CR network may also facilitate a rostrocaudal spread of activation. Together, these observations extend on the current view of excitatory interneurons in spinal sensory processing, which ascribes a relative limited role of linking low threshold modality (innocuous) tactile input to excite more dorsal nociceptive circuitry (*Duan et al., 2014*; *Peirs and Seal, 2016*; *Yu et al., 2017*). We suggest these neurons also provide a polysynaptic network to enhance/amplify local excitation, prime the region for subsequent responses, and spread excitatory signaling across modality borders.

Though not directly assessed, the cellular mechanisms that underpin these observations likely relate to a number of properties in the interconnected CR network. Specifically, CR neurons in this region, as well as the majority of their targets, had an excitatory phenotype and were coupled by excitatory synapses (*Figure 2 and 4*). Thus, strong activation of this network likely established reverberatory activity among these circuits as proposed by others (*Hachisuka et al., 2018*), as well as potentially inducing short term plasticity at these interconnected synapses. In vitro recordings of enhanced synaptic signaling in CR neurons and LI projection neurons following photostimulation (*Figure 3 and 5*) certainly support this explanation, however, future work will be required to assess the relative importance of these observations to extended behavioural responses.

## Conclusions

The results from this study confirm that CR neurons form an excitatory dorsal horn network that contributes to spinal pain signaling and can amplify pain signals in the absence of peripheral input. The strong interconnectivity of this neurochemially-defined subpopulation in the dorsal horn means they are ideally positioned to alter incoming sensory information prior to its relay to higher brain centers. This capacity is clearly demonstrated in the nocifensive responses elicited by spinal photostimulation. Importantly, however, our work has largely focused on characterizing CR microcircuits and establishing their functional roles in sensory experience by experimental activation using optogenetics. Future work must continue to determine how these circuits respond during peripherally evoked sensory processing, as we have here using halorhodopsin. Finally, our data in SNI animals suggest CR networks change under neuropathic conditions to become more excitable (*Figure 8E–H*). The question of how pathology and injury can recruit or alter these CR circuits will also be critical for determining how they contribute to symptoms of pathological pain, and how best to target them for therapeutic benefit.

## Materials and methods

### Key resources table

| Reagent type (species) or resource | Designation | Source or reference | Identifiers | Additional information |
|---|---|---|---|---|
| Antibody | Goat anti-calretinin | SWANT, Bellinoza, Switzerland | Cat# ABS571, RRID: AB_10000342 | IF(1:500) |
| Antibody | Chicken anti-GFP (polyclonal antibody) | Abcam, UK | Cat# ab15051, RRID: AB_300798 | IF(1:1000) |

*Continued on next page*

*Continued*

| Reagent type (species) or resource | Designation | Source or reference | Identifiers | Additional information |
|---|---|---|---|---|
| Antibody | Rabbit anti-GFP (polyclonal antibody) | Frontier Institute Co. Ltd, Japan | Cat# GFP-Rb-Af2020; RRID: AB_2571573 | IF(1:5000) |
| Antibody | Goat anti-Homer1 (polyclonal antibody) | Frontier Institute Co. Ltd, Japan | Cat# Homer1-Go, RRID:AB_2631104 | IF(1:1000) |
| Antibody | Rat anti-mCherry (monoclonal antibody) | Thermo Fisher Scientific | Cat# M11217, RRID:AB_2536611 | IF(1:1000) |
| Antibody | Rabbit anti-NK1R | Sigma-Aldrich | Cat# S8305, RRID:AB_261562 | IF(1:2000) |
| Antibody | Rabbit anti-Pax2 (polyclonal antibody) | Thermo Fisher Scientific | Cat# 71–6000, RRID:AB_2533990 | IF(1:1000) |
| Antibody | Rabbit anti-PKCγ (polyclonal antibody) | Santa Cruz Biotechnology | Cat# sc-211, RRID:AB_632234 | IF(1:1000) |
| Antibody | Guinea Pig anti-somatostatin | Dr Philippe Ciofi | Philippe Ciofi, INSERM U1215. Bordeaux, France. | IF(1:1000) |
| Antibody | Mouse anti-VGLUT2 (monoclonal antibody) | Millipore, Chemicon International, UK | Cat# MAB5504, RRID:AB_2187552 | IF(1:500) |
| Antibody | Rat anti-substance P | Oxford Biotech, UK | Cat No: OBT0643S | IF(1:200) |
| Antibody | Rabbit anti-CGRP | Sigma-Aldrich | Cat# C8198, RRID:AB_259091 | IF(1:10000) |
| Antibody | Mouse anti-NF200 (monoclonal antibody) | Sigma-Aldrich | Cat# N0142, RRID:AB_477257 | IF(1:500) |
| Antibody | Guinea Pig anti-VGLUT1 (polyclonal antibody) | Millipore (Chemicon) | Cat# AB5905, RRID:AB_2301751 | IF(1:5000) |
| Antibody | Guinea Pig anti-VGLUT2 (polyclonal antibody) | Millipore (Chemicon) | Cat# AB2251-I, RRID:AB_2665454 | IF(1:5000) |
| Antibody | Rabbit anti-VGlut2 (polyclonal antibody) | Synaptic Systems | Cat# 135 402, RRID:AB_2187539 | IF(1:5000) |
| Antibody | Guinea Pig anti-VGLUT3 (polyclonal antibody) | Frontier Institute Co. Ltd, Japan | Cat# VGluT3-GP-Af920, RRID:AB_2571856 | |
| Antibody | Chicken anti-prostatic acid phosphatase (polyclonal antibody) | Aves Labs. Inc, OR, USA | Cat# PAP, RRID:AB_2313557 | IF(1:1000) |
| Genetic reagent (virus) | AAV9-CB7-Cl ~ mCherry | addgene, USA | Cat#105544-AAV9 | |
| Genetic reagent (M. musculus) | Calb2-IRES-cre: B6(Cg)-Calb2tm1(cre)Zjh/J | The Jackson Laboratories, USA | Cat# JAX:010774, RRID:IMSR_JAX:010774 | |
| Genetic reagent (M. musculus) | Ai32:B6;129S-Gt(ROSA)26Sortm32(CAG-COP4*H134R/EYFP)Hze/J | The Jackson Laboratories, USA | Cat# JAX:012569, RRID:IMSR_JAX:012569 | |
| Genetic reagent (M. musculus) | Ai34: B6;129S - Gt(ROSA)26Sortm34.1(CAG-Syp/tdTomato)Hze/J | The Jackson Laboratories, USA | JAX:012570 RRID:IMSR_JAX:012570 | |
| Genetic reagent (M. musculus) | Ai39:B6;129S-Gt(ROSA)26Sortm39(CAG-hop/EYFP)Hze/J | The Jackson Laboratories, USA | JAX:014539, RRID:IMSR_JAX:014539 | |
| Genetic reagent (M. musculus) | CReGFP | Prof Hannah Monyer, Heidelberg University, Germany | | |
| Genetic reagent (M. musculus) | Advillin eGFP | GENSAT, MMRRC | Cat# 034769-UCD, RRID:MMRRC_034769-UCD | |

*Continued on next page*

Continued

| Reagent type (species) or resource | Designation | Source or reference | Identifiers | Additional information |
|---|---|---|---|---|
| Software, algorithm | Neurolucida for Confocal Software | MBF Bioscience, VT, USA | https://www.mbfbioscience.com/neurolucida | |
| Software, algorithm | Neurolucida Explorer | MBF Bioscience, VT, USA | https://www.mbfbioscience.com/neurolucidaexplorer | |
| Software, algorithm | AxoGraph X | Dr. John Clements | https://axograph.com/ | |
| Software, algorithm | Zen Black | Carl Zeiss, Germany | https://www.zeiss.com/microscopy/int/products/microscope-software/zen.html | |
| Software, algorithm | JWatcher | Daniel T. Blumstein et al., University of California Los Angeles and Macquarie University, Sydney | http://www.jwatcher.ucla.edu | |

## Animals and ethics

Optogenetic studies were carried out on mice derived by crossing *Calb2*-IRES-cre (Jackson Laboratories, Bar Harbor, USA; #010774) with either Ai32 (Jackson Laboratories, Bar Harbor, USA; #024109) or Ai39 (Jackson Laboratories, Bar Harbor, USA, #014539) to generate offspring where ChR2/YFP or NpHR/YFP was expressed in CR$^+$ cells (CR$^{Cre}$;Ai32 or CR$^{Cre}$;Ai39). Axon terminal labelling experiments crossed Calb2-IRES-cre and Ai34D (Jackson Laboratories, Bar Harbour, USA, # 012570) mice to generate offspring with CR$^+$ axon terminals labelled with TdTomato. In control experiments, another transgenic mouse line with enhanced green fluorescent protein expressed under the control of the *calretinin* (*Calb2*) promoter (CReGFP) was used (*Caputi et al., 2009*). All experimental procedures were performed in accordance with the University of Newcastle's animal care and ethics committee (protocols A-2013–312 and A-2016–603). Animals of both sexes were used for electrophysiology (age: 3–12 months) and behavior experiments (age: 8–12 weeks).

## Spinal slice preparation

Acute spinal cord slices were prepared using previously described methods (*Graham et al., 2003*; *Graham et al., 2011*). Briefly, animals were anaesthetized with ketamine (100 mg/kg i.p) and decapitated. The ventral surface of the vertebral column was exposed and the spinal cord rapidly dissected in ice-cold sucrose substituted cerebrospinal fluid (ACSF) containing (in mM): 250 sucrose, 25 NaHCO$_3$, 10 glucose, 2.5 KCl, 1 NaH$_2$PO$_4$, 1 MgCl and 2.5 CaCl$_2$. Either parasagittal or transverse slices were prepared (lumbar segments L1 to 5, 200 µm thick: LI-L5 300 µm thick, respectively) both using a vibrating microtome (Campden Instruments 7000 smz, Loughborough, UK). Targeted recordings from both YFP-expressing and unidentified cells were undertaken in parasagittal slices, whereas targeted PN recordings used slices in the transverse plane. Slices were transferred to an interface incubation chamber containing oxygenated ACSF (118 mM NaCl substituted for sucrose) and allowed to equilibrate at room temperature for at least one hour prior to recording.

## Patch clamp electrophysiology

Following incubation, slices were transferred to a recording chamber and continuously superfused with ACSF bubbled with carbanox (95% O$_2$, 5% CO$_2$) to achieve a final pH of 7.3–7.4. All recordings were made at room temperature. Neurons were visualised using a 40x objective and near-IR differential interference contrast optics. To identify CR-ChR2/CR-NpHR$^+$ neurons, which expressed YFP, slices were viewed under fluorescence using a FITC filter set. CR$^+$ neurons were concentrated within lamina II of the dorsal horn, as described previously (*Smith et al., 2015*; *Smith et al., 2016*), and is easily identified as a plexus of YFP fibres and soma under fluorescent microscopy. All recordings were made either within or dorsal to this YFP plexus. The parasagittal slicing approach allowed easy differentiation of the two CR populations we have previously described (*Smith et al., 2015*;

*Smith et al., 2016*). Specifically, the CR excitatory population exhibits a restricted dendritic profile, whereas less common inhibitory CR neurons possess extensive rostro-caudal projecting dendritic arbours. Patch pipettes (4–8 MΩ) were filled with either a potassium gluconate based internal for recordings of excitatory input and action potential (AP) discharge, containing (in mM): 135 $C_6H_{11}KO_7$, 6 NaCl, 2 $MgCl_2$, 10 HEPES, 0.1 EGTA, 2 MgATP, 0.3 NaGTP, pH 7.3 (with KOH); or a caesium chloride-based internal solution for inhibitory input recordings, containing (in mM): 130 CsCl, 10 HEPES, 10 EGTA, 1 $MgCl_2$, 2 MgATP and 0.3 NaGTP, pH 7.35 (with CsOH). Neurobiotin (0.2%) was included in all internal solutions for *post-hoc* cell morphology. All data were acquired using a Multiclamp 700B amplifier (Molecular Devices, Sunnyvale, CA, USA), digitized online (sampled at 10–20 kHz, filtered at 5–10 kHz) using an ITC-18 computer interface (Instrutech, Long Island, NY, USA) and stored using Axograph X software (Molecular Devices, Sunnyvale, CA, USA).

AP discharge patterns were assessed in current clamp from a membrane potential of ~ −60 mV by delivering a series of depolarising current steps (1 s duration, 20 pA increments). AP discharge was classified using previously described criteria (*Graham et al., 2004*; *Graham et al., 2007*). Briefly, delayed firing (DF) neurons exhibited a clear interval between current injection and the onset of the first AP; tonic firing (TF) neurons exhibited continuous repetitive AP discharge for the duration of the current injection; initial bursting (IB) neurons were characterised by a burst of AP discharge at the onset of the current injection; and single spiking (SS) neurons only fired a single AP at the beginning of the current step. Input resistance and series resistance were monitored throughout all recordings and excluded if either of these values changed by more than 10%. No adjustments were made for liquid junction potential. The subthreshold currents underlying AP discharge were assessed using a voltage-clamp protocol that delivered a hyperpolarizing step to −100 mV (1 s duration) followed by a depolarizing step to −40 mV (200 ms duration) from a holding potential of −70 mV. This protocol identifies four major ionic currents previously described in dorsal horn neurons, including the outward potassium currents (rapid and slow $I_A$) and the inward currents, T-type calcium and non-specific cationic current $I_h$.

## In vitro optogenetics

Photostimulation was achieved using a high intensity LED light source (CoolLED pE-2, Andover, UK) delivered through the microscopes optical path and controlled by Axograph X software. Recordings from excitatory versus inhibitory ChR2-YFP neurons were distinguished using their morphology in the parasagittal slice and distinct electrophysiological profiles (*Smith et al., 2015*; *Smith et al., 2016*). Photocurrents were first characterised in ChR2-YFP neurons using a current versus light intensity (488 nm, 1 s duration) analysis in voltage clamp mode. Combinations of neutral density filters were used to reduce photostimulation intensity (0.039–16 mW). To assess the ability and reliability of photostimulation to evoke AP discharge in ChR2-YFP neurons the recording mode was switched to current clamp and brief photostimuli (16 mW, 1 ms) were delivered at multiple frequencies 5 Hz, 10 Hz and 20 Hz (1 s duration). We then used channelrhodopsin-2 assisted circuit mapping (CRACM) to characterise the connectivity of ChR2 neurons within the dorsal horn. The postsynaptic circuits receiving input from ChR2-YFP neurons were characterized by delivering photostimulation (16 mW, 1 ms) every 12 s during patch clamp recordings and assessing current responses for photostimulation associated synaptic input. These recordings were made from ChR2-YFP neurons as well as 3 populations of neurons that did not show YFP expression. These populations were classified relative to the distinct YFP plexus within lamina II as either: 1) Plexus - within the YFP plexus; 2) Dorsal - dorsal to YFP plexus; or 3) Projection neurons - dorsal to the YFP plexus and retrogradely labelled (see below). The response of these populations was also assessed during prolonged photostimulation using two stimulus paradigms: 1) a 2 s continuous photostimulation (16 mW); and 2) a 1 s photostimulation (16 mW, 1 ms pulses @ 10 Hz for 1 s).

## Projection neuron recordings

To identify lamina I projection neurons in slice recording experiments, a subset of animals (n = 2) underwent surgery to inject a viral tracer, specifically AAV9.CB7.CI.mCherry (viral titre = 2.5×$10^{13}$ vg/mL), into the parabrachial nucleus (PBN) to retrogradely-label lamina I PNs for subsequent targeted patch-clamp recording experiments. Briefly, mice were anaesthetised with isoflurane (5% induction, 1.5–2% maintenance) and secured in a stereotaxic frame (Harvard Apparatus,

Massachusetts, USA). A small craniotomy was performed and up to 700 nL of the viral sample was injected using a picospritzer (PV820, WPI, Florida, USA) into the PBN bilaterally. These injections were made 5.25 mm posterior to bregma, ± 1.2 mm of midline and 3.8 mm deep from skull surface, using coordinates refined from those in the mouse brain atlas (*Paxinos and Franklin, 2001*). Injections were made over 5 min and the pipette left in place for a further 7–10 min to avoid drawing the virus sample along the pipette track. Animals were allowed to recover for 3 weeks to allow sufficient retrograde labelling of projection neurons before spinal cord slices were prepared. CRACM was then performed as above for other DH populations. The brain from each animal was also isolated and brainstem slices containing the PBN were prepared to confirm the injection site, which was appropriately focussed on PBN in all cases. Spinal cord slices were obtained using methods described above (*spinal slice preparation*) and mCherry positive neurons were visualised for recording using a Texas Red filter set.

## Patch clamp data analysis

All electrophysiology data were analysed offline using Axograph X software. ChR2 photocurrent amplitudes were measured as the difference between baseline and the steady state portion of the photocurrent. Excitatory and inhibitory photostimulation-evoked synaptic currents elicited by brief photostimulation, hereafter termed optical postsynaptic currents (oEPSCs and oIPSCs), were captured episodically and averaged (10 trials). Peak amplitude, rise time (10–90% of peak) and decay time constant (10–90% of the decay phase) were measured from average oEPSCs and oIPSCs. Response latency was also measured on averaged records, as the time between the onset of photostimulation and onset of the oEPSC/oIPSC. In photostimulation responses that contained multiple components a semi-automated peak detection procedure was used to determine the latency of all responses. To differentiate direct (monosynaptic) input from indirect (polysynaptic) response components the photostimulation recruitment time for CR-ChR2[+] neurons was determined as the latency between the onset of photostimulation and the onset of AP discharge. In addition, the average time between spiking in a presynaptic neurons and a monosynaptic response in synaptically connected neurons was taken from previous paired recording studies in the spinal dorsal horn (*Santos et al., 2007*; *Lu and Perl, 2003*; *Lu and Perl, 2005*). These data account for the combination of AP conduction and synaptic delay that takes ~2.5 ms. Thus, windows were set for oEPSCs and oIPSCs to be considered monosynaptic by adding the photostimulation recruitment time, conduction and synaptic delays (±2 standard deviations) of photostimulation recruitment time. Responses outside these windows were considered to more likely arise from polysynaptic activity. For longer photostimulation paradigms both oEPSCs and spontaneous excitatory postsynaptic currents (sEPSCs) were detected using a sliding template method (a semi-automated procedure in the Axograph package). Average oEPSC/sEPSCs frequency was calculated over 100 ms epochs by multiplying the number of events in each epoch by 10.

To isolate oEPSCs and oIPSCs in CR-ChR2 neurons, the photocurrents were first subtracted using a pharmacological approach. For oEPSCs (K[+] gluconate-based internal), photocurrents were isolated following application of CNQX (10 µM) and then scaled to the peak photocurrent before drug application. The isolated photocurrent was then subtracted from the pre-CNQX traces leaving the isolated synaptic response (*Figure 2—figure supplement 1*). The same procedure was repeated for oIPSCs (CsCl-based internal solution), except responses were obtained under three conditions following sequential application of CNQX (10 µM), bicuculline (10 µM), and strychnine (1 µM). In this case, the isolated photocurrent (recorded in CNQX, bicuculline and strychnine) was subtracted from the photostimulation responses under each drug condition (*Figure 2—figure supplement 2*).

## Optogenetic stimulation for Fos activation mapping

The postsynaptic circuits targeted by CR[+] neurons were assessed by delivering spinal photostimulation to anaesthetised CR[Cre];Ai32 animals (and CReGFP control animals) and then processing spinal cords for Fos-protein and a range of additional neurochemical markers. Animals (n = 5) were anaesthetised with isoflurane (5% initial, 1.5–2% maintenance) and secured in a stereotaxic frame. A longitudinal incision was made over the T10-L1 vertebrae and a laminectomy was performed on the T13 vertebra. Unilateral photostimulation (10 mW, 10 ms pulses @ 10 Hz for 10 min) was then delivered to the exposed spinal cord by positioning an optic fiber probe (400 nm core, 1 mm fiber length,

Thor Labs, New Jersey, U.S.A) above the spinal cord surface using the stereotaxic frame. Photostimulation was delivered by a high intensity LED light source attached to the probe via a patch cord. Following photostimulation animals remained under anaesthesia for a further 2 hr for subsequent comparison of Fos expression in neurochemically defined DH neurons. Animals were then anaesthetised with ketamine (100 mg/kg i.p) and perfused transcardially with saline followed by 4% depolymerised formaldehyde in 0.1M phosphate buffer. Sections from L3 and L4 spinal segments were processed for immunocytochemistry by incubating in a cocktail of antibodies including chicken anti-GFP and goat anti-cFos, with either rabbit anti-NK1R, rabbit anti-somatostatin, rabbit anti-PKCγ or rabbit anti-Pax2, as described previously (*Hughes et al., 2013*). Full details of primary antibodies are provided in the Key Resources Table. Primary antibody labelling was detected using species-specific secondary antibodies conjugated to rhodamine (diluted 1:100), Alexa 488, Alexa 647 (diluted 1:500). NK1-immunolabelling was visualised using a biotinylated anti-rabbit antibody (diluted 1:500) followed by a Tyramide signal amplification step using a tetramethylrhodamine kit (PerkinElmer Life Sciences, Boston, MA, USA), as described previously (*Hughes et al., 2013*). Sections were incubated in primary antibodies for 72 hr and in secondary antibodies for 12–18 hr. All primary and secondary antibody cocktails were made up in 0.1M phosphate buffer with 0.3M NaCl and 0.3% Triton X-100. All secondary antibodies were purchased from Jackson Immunoresearch, West Grove, PA, USA.

## Transgenic axon terminal labelling

Analysis of CR$^+$ neuron input to putative projection neurons was undertaken in tissue from CR$^{Cre}$; Ai34 mice that selectively labelled CR$^+$ axon terminals with tdTomato. Animals were anaesthetised with sodium pentobarbitone (30 mg/kg *i.p.*) and perfused transcardially with Ringer solution followed by 4% depolymerised formaldehyde in 0.1M phosphate buffer. Sections from L3 and L4 spinal segments were processed for immunohistochemistry by incubating in cocktails of antibodies including chicken anti-GFP, goat anti-calretinin, goat anti-Homer1, rat anti-mCherry, rabbit anti-NK1R, guinea pig anti-somatostatin, and mouse anti-VGLUT2. For full details of these antibodies, see Key resources table. Primary antibody labelling was detected using species-specific secondary antibodies conjugated to rhodamine, Alexa 488, Alexa 647 (Jackson Immunoresearch, West Grove, PA, USA).

## Calretinin inputs onto filled Projection Neurons

As noted above, 0.2% Neurobiotin was included in all internal recording solutions to recover recorded cell morphology. In the AAV-mediated targeted recordings of LI projection neurons, the cells were recovered using a streptavidin ~Cy5 secondary antibody before initial imaging (z = 1 μm, scan speed 400, pinhole 1AU) using a water immersion 25x objective on a Leica TCS SP8 scanning confocal microscope equipped with Argon (458, 488, 514 nm), DPSS (561 nm) and HeNe (633) lasers. Slices that contained recovered PNs were reacted with chicken anti-GFP (see Table 2 for details), to resolve axon boutons of CR neurons in close apposition with labelled PNs. Spinal slices were re-sectioned to 50 μm thickness, mounted in glycerol and imaged using both a 40x oil and 63x water immersion objective. Boutons were identified as rounded YFP-labelled profiles directly apposed to labelled PN dendrites.

## Optogenetic probe surgery

Animals were anaesthetised with isoflurane (5% initial, 1.5–2% maintenance) shaved over the thoracolumbar vertebral column, secured in a stereotaxic frame and the surgical site was cleaned with chlorhexadine. Using aseptic procedures, a 3 cm incision was made over the T10-L1 vertebrae and paraspinal musculature removed. The intervertebral space between T12 and T13 was cleared to expose the spinal cord and overlying dura. Surgical staples were attached to the corresponding T12 and T13 to provide a rigid fixation point of attachment for the fiber optic probe. A probe (400 nm core, 1 mm fiber length, Thor Labs, New Jersey, U.S.A) was then positioned over the exposed spinal cord, lateral to the midline, and fixed in place between the vertebrae and associated surgical staples using orthodontic crown and bridge cement (Densply, Woodbridge, Canada). The surgical site was closed with sutures and surgical staples, and the animals were allowed to recover before being returned to their home cage for 7 days before spinal in vivo photostimulation.

## In vivo photostimulation and behaviour

Animals were briefly anaesthetised (isoflurane, 5%) to attach a fiber optic patch cord (400 nm core, Thor Labs) to the implanted fiber optic probe before being placed in a small Perspex testing cylinder (10 cm diameter, 30 cm height) and allowed to habituate for 30 mins in the three days preceding photostimulation and for 20 min prior to photostimulation. The patch cord was attached to a high intensity LED light source (DC2100, 470 nm, Thor Labs) and photostimulation (10 mW, 10 ms pulses @ 10 Hz for 10 s) was delivered. Behavioural responses were recorded using a Panasonic video camera (Pansonic HC-V770M, Panasonic, Kadoma, Japan). In all experiments, animals were first introduced and acclimatised to the testing chamber for 3 days prior to testing, and video recordings captured 5 min of behaviour before and after photostimulation. Behavioural experiments were performed and analysed blind to animal genotype and/or experimental conditions. Animal status was coded and then maintained confidential until the completion of experiments and analysis.

In some experiments the testing conditions were altered to address specific aspects of the photostimulation response. The relationship between stimulation intensity and behaviour was assessed in a subset of animals (n = 6) received varying photostimulation intensities (0.5–20 mW, 10 ms pulses @ 10 Hz for 10 s), with a 30 min break between each stimulus. The potential for a peripheral signalling component to photostimulation responses was tested by characterizing baseline (control) photostimulation responses (n = 4), and then repeating photostimlation in these animals following local anaesthesia of the hindpaw (2% lidocaine, intraplantar), and vehicle (saline, intraplantar) injection. To assess the functional consequences of repeated spinal photostimulation, animals (n = 9) received 2 photostimuli delivered two mins apart. To test the nociceptive nature of spinal photostimulation, animals (n = 5) were administered morphine 30 min prior to photostimulation. Three morphine doses were assessed (3, 10 and 30 mg/kg, s.c), as well as a saline vehicle control. Animals first underwent two photostimulation intensities (10 and 20 mW) with no morphine to determine baseline responses. Drug treatments were randomly assigned such that each animal received all concentrations and a 48 hr interval between each drug administration allowed morphine washout.

In experiments to assess the activation of higher order brain regions in response to photostimulation, animals (n = 5 CR[Cre];Ai32, n = 5 CReGFP) were placed in a testing chamber (30 cm length, 25 cm width and 40 cm height) with food and water available ad libitum for 6 hr to eliminate any Fos activation caused by handling, the environment, or anaesthesia. Animals received photostimulation (10 mW, 10 ms pulses @ 10 Hz for 10 s) before being left for a further 2 hr, to allow development of Fos expression, and then perfused transcardially with 4% PFA. Brains were dissected and post-fixed in 4% PFA overnight then stored in 30% sucrose. Serial sections were cut from the forebrain (40 μm) and brainstem (50 μm) using a freezing microtome (Leica Microsystems, SM2000R) and a 1 in four series were processed for Fos protein labelling (1:5000, rabbit polyclonal. Santa Cruz Biotechnology, CA, USA). Fos positive cells were then manually counted from cingulate, insula, primary somatosensory cortex (hindpaw), and parabrachial nucleus. Reference images from the Allen Mouse Brain atlas were used to define the boundaries of each structure for counts (2011 Allen Institute for Brain Science. Allen Adult Mouse Brain Atlas. Available from: mouse.brain-map.org). Counts were made on four sections from each CR[Cre];Ai32 and CReGFP mouse (cingulate, insula, somatosensory cortex, PBN) taken from the side contralateral to spinal photostimulation (8.5X magnification).

In photo-inhibition experiments the fiber optic patch cord was attached to CR-NpHR animals (n = 8) which were then acclimatised to the testing tube as described previously. The patch cord was attached to the same laser described in in vitro methods. The simplified up down method (SUDO) (*Bonin et al., 2014*) of von Frey testing was used to establish mechanical withdrawal thresholds both with and without photo-inhibition. Animal were habituated in the testing chamber for 30 min for the 3 days prior to testing. Over four days of testing the von Frey threshold testing was assessed in each animal once with and once without photo-inhibition in alternating order. Withdrawal scores averaged across the four trial days and converted to withdrawal threshold in grams (*Bonin et al., 2014*).

All in vivo photostimulation-induced behaviour was analysed using JWatcher v1.0 event recorder (*Blumstein and Daniel, 2007*). Behavioural responses were encoded from the video recordings of photostimulation including 5 min pre- and post-photostimulation, played back at half-speed (30 fps). All behaviours targeted at the left or right hind limbs and the midline were coded. In a subset of videos coding was expanded to differentiate left/right paw and leg, as well as back and tail. The

duration of all targeted behaviours was then binned (time epochs) and converted to a colour scale showing the proportion of epoch spent in specific behaviours for visualization.

## Mouse model of neuropathic pain

To study spinal circuitry in allodynic animals, we carried out the spared nerve injury model (SNI; *Decosterd and Woolf, 2000*) in CR[Cre];Ai32 mice (n = 4). Specifically, a 2 to 3 mm length of the tibial and common peroneal nerves was removed between two tight ligatures with 7–0 Mersilk under general anesthesia, with the sural nerve left intact and not manipulated (*Boyle et al., 2019*). Behavioral responses to mechanical stimulation of skin regions innervated by the sural nerve were tested using von Frey filaments with logarithmically incremental stiffness prior to, and following surgery. The 50% paw withdrawal threshold was calculated using the SUDO method (*Bonin et al., 2014*).

## Conditioned place aversion testing

Conditioned place aversion (CPA) testing was used to assess the aversive nature of CR-ChR2 photostimulation. The CPA apparatus consisted of a two-chamber black perspex box (50 cm length, 25 cm width and 50 cm height) with a divider allowing free access to each chamber. To differentiate the two chambers one side contained cross-hatched markings using tape on the floor and crosses on the walls. On the first experimental day each animals baseline preference was determined. The optic patch cord was attached (as described above) and animal placed in the centre of the CPA apparatus. Animals were allowed to freely move between both chambers for 10 mins, prior to commencement of data collection. The chamber where an animal spent the most time was deemed the preferred side, and subsequently designated 'photostimulation on' while the non-preferred side was designated 'photostimulation off'. Animals then underwent 4 × 20 min trials over 2 days (morning and afternoon), with the condition that entry into the 'photostimulation on' chamber triggered photostimulation (10 mW, 10 ms pulses @ 10 Hz for 10 s in every minute ie. 10 s on, 50 s off) until the animal returned to the 'photostimulation off' chamber. Following the CPA trials, the persistence of the CPA memory was assessed via short term (1 hr post testing - STM) and long term tests (24 hr post testing - LTM). In these tests animals were allowed to freely move around the CPA apparatus for 10 mins with no photostimulation. All trials and tests were captured from above the CPA apparatus (via the video camera), digitized, then analysed using semi-automated behavioural tracking procedures within Ethovision software (Noldus Information Technology, Wageningen, Netherlands).

## Statistical analysis

All data are presented as mean ± the standard error of the mean (SEM) unless otherwise stated. Shapiro-Wilk's test determined if data were normally distributed. For normally distributed data one-way ANOVAs were performed with a student Newman-Keuls *post-hoc* test to compare oEPSC and oIPSC properties between neuron groups and for all behaviour analyses. Non-normally distributed data was compared using the Kruskal Wallis test with Wilcoxon–Mann–Whitney *post-hoc* testing. Paired t-tests compared sEPSC frequency before and after photostimulation in CR[+] neuron and projection neuron populations.

## Acknowledgements

We thank Dr Philippe Ciofi for the guinea pig anti-somatostatin primary antibody, and both Christine Watt and Robert Kerr for expert technical assistance. This work was funded by the National Health and Medical Research Council (NHMRC) of Australia (grants 631000 and 1043933 to BAG, 1067146 to RJC and 1125478 to CVD), the Biotechnology and Biological Sciences Research Council (BBSRC) of the United Kingdom (grant BB/J000620/1 and BB/P007996/1 to DIH), and the Hunter Medical Research Institute (BAG and RJC).

# Additional information

## Funding

| Funder | Grant reference number | Author |
|---|---|---|
| National Health and Medical Research Council | 631000 | Brett A Graham |
| National Health and Medical Research Council | 1043933 | Brett A Graham |
| Biotechnology and Biological Sciences Research Council | BB/J000620/1 | David I Hughes |
| Biotechnology and Biological Sciences Research Council | BB/P007996/1 | David I Hughes |
| National Health and Medical Research Council | 1067146 | Robert J Callister |
| National Health and Medical Research Council | 1125478 | Christopher V Dayas |
| Hunter Medical Research Institute | | Robert J Callister Brett A Graham |

The funders had no role in study design, data collection and interpretation, or the decision to submit the work for publication.

## Author contributions

Kelly M Smith, Conceptualization, Formal analysis, Investigation, Methodology, Writing—original draft, Writing—review and editing; Tyler J Browne, Formal analysis, Investigation, Methodology, Writing—original draft, Writing—review and editing; Olivia C Davis, A Coyle, Formal analysis, Investigation, Writing—review and editing; Kieran A Boyle, Formal analysis, Supervision, Investigation, Methodology, Writing—review and editing; Masahiko Watanabe, Resources, Writing—review and editing; Sally A Dickinson, Jacqueline A Iredale, Mark A Gradwell, Formal analysis, Writing—review and editing; Phillip Jobling, Formal analysis, Supervision, Methodology, Writing—review and editing; Robert J Callister, Christopher V Dayas, Conceptualization, Supervision, Funding acquisition, Methodology, Writing—review and editing; David I Hughes, Brett A Graham, Conceptualization, Formal analysis, Supervision, Funding acquisition, Investigation, Methodology, Writing—original draft, Project administration, Writing—review and editing

## Author ORCIDs

Kelly M Smith https://orcid.org/0000-0002-3039-5002
Olivia C Davis https://orcid.org/0000-0001-8792-7324
Masahiko Watanabe http://orcid.org/0000-0001-5037-7138
David I Hughes https://orcid.org/0000-0003-1260-3362
Brett A Graham https://orcid.org/0000-0002-8070-0503

## Ethics

Animal experimentation: All studies carried out in Glasgow were in accordance with the European Community directive 86/609/EEC and UK Animals (Scientific Procedures) Act 1986. All studies carried out at University of Newcastle were in accordance with the Animal Research Act 1985 (NSW), under the guidelines of the National Health and Medical Research Council Code for the Care and Use of Animals for Scientific Purposes in Australia (2013). All animal handling and experimental procedures were performed under approved institutional animal care and ethics committee protocols (University of Newcastle: A-2013-312 and A2016-603; University of Glasgow).

## Decision letter and Author response

Decision letter https://doi.org/10.7554/eLife.49190.sa1
Author response https://doi.org/10.7554/eLife.49190.sa2

## Additional files

### Supplementary files
• Transparent reporting form

### Data availability
All data generated or analysed during this study are included in the manuscript and supporting files.

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
