## [Decision Letter]

**Acceptance summary:**

In this manuscript the authors present a very careful and multidisciplinary analysis and show of the calretinin (CR^+^) population of dorsal horn interneurons. Previous studies disagreed as to the contribution. One group concluded that the interneurons mediate innocuous mechanical stimulation; another group concluded that they are nociresponsive. Here the authors establish that the CR^+^ neurons do, in fact, mediate a strong nociceptive mechanical sensation, that the neurons transmit information directly to nociceptive projection neurons of the dorsal horn and interestingly contribute to a process through which excitation on interneurons is amplified and allowing information to transfer cross borders.

**Decision letter after peer review:**

Thank you for submitting your article "Calretinin positive neurons form an excitatory amplifier network in the spinal cord dorsal horn" for consideration by *eLife*. Your article has been reviewed by three peer reviewers, one of whom is a member of our Board of Reviewing Editors, and the evaluation has been overseen by Ronald Calabrese as the Senior Editor. The reviewers have opted to remain anonymous.

The reviewers have discussed the reviews with one another and the Reviewing Editor has drafted this decision to help you prepare a revised submission. There is considerable agreement that your manuscript is beautifully executed and provides considerable new information of great interest to the field. However, there are several significant concerns that need to be addressed in a revised manuscript before publication in *eLife* could be considered.

First, there was some disappointment noted that the study only analyzed mechanical pain, in an acute setting. Thus, in a revision, it is essential that the effect of DTR-mediated deletion (or optogenetic-mediated inhibition) of the CR neurons be tested in a model of persistent pain. Including information about heat responsiveness would be ideal, but that is not essential. If not included, its absence should be discussed.

Lastly, very significant concern was raised as to the possibility that dorsal horn CR interneurons are not the sole contributors to your findings. If indeed a population of CR expressing sensory neurons contributes to the phenotype that you described (for example, secondary to activation of their terminals in the spinal cord), then the interpretation that you have drawn would have to be considerably tempered. In a revision, you should include results of an examination of the DRG for CR reporter expression. We appreciate that you may not have access to mice that would permit an intersectional approach with which you could restrict the analysis to the dorsal horn neurons. For this reason, we ask that, at the minimum, and depending on the results of the reporter analysis in the DRG, the Discussion section (and elsewhere where appropriate) should address the concern.

*Reviewer 1:*

This is an interesting and incredibly detailed paper that addresses the contribution of calretinin neurons (presumably interneurons) in pain processing. The authors use a wealth of approaches and demonstrate quite convincingly that the great majority are excitatory, that they arborize extensively in superficial dorsal horn, that they engage many other CR neurons as well as non CR neurons, including some NK1 receptor expression neurons. The latter are presumed to be projection neurons. Based on electrophysiological and some behavioral analyses involving optogenetic activation of these neurons the authors conclude that the CR interneurons drive pain behavior by an excitatory action (including an interesting feed forward amplification that appears to prolong licking/biting) through these networks. The study adds considerably to many recent reports using somewhat comparable approaches that have studied the contribution of different interneuron populations to pain and itch processing.

My concern reflects several issues, some of which might be addressable in a revision, but would require additional work.

1) There are CR-expressing sensory neurons. For example, the Ernfors data base, finds expression of the gene in myelinated afferents (including proprioceptors). Although these afferents likely terminate in deep dorsal horn, it is always possible that there is some expression in afferents that terminate in superficial dorsal horn. That needs to be ruled out. At the minimum, tomato expression in sensory neurons should be reported.

2) The cross that generated the pattern of expression was not tamoxifen-dependent. In other words, the pattern reflects embryonic Cre recombination. Can the authors be certain that they are dealing with the true CR pattern in the adult?

3) The authors only looked at mechanical stimulation. Is there any change in heat pain responsiveness? What about itch?

4) Most importantly, the authors only looked at acute pain behaviors provoked by activation of a large population of the CR neurons. It would be of great interest to understand their contribution, if any, in the setting of, for example, a model of neuropathic pain (partial sciatic nerve injury). Does DTR-mediated killing of cells, or inhibition of the cells optogenetically using the approaches described in the paper, alter mechanical hypersensitivity provoked in the setting of injury?

5) Related to the latter question: What types of afferents engage the cells? Of particular interest are the Aβ afferents, which may carry the input that underlies mechanical allodynia in the setting of nerve injury.

*Reviewer 2:*

The authors study the contribution of the Calretinin populations of interneurons within the dorsal horn using a Cre driver mouse line combining spinal electrophysiology, neuroanatomy combined optogenetics and behavioral assessment. There have been previous studies of this population of interneurons I think the advances are a more detailed assessment of the local circuitry of these interneurons and demonstration of a significant input onto lamina 1 projection neurons as well as the conclusion that although there is a small population of inhibitory CR interneurons the majority are excitatory and highly inter-connected acting in a feedforward manner to enhance excitatory signalling. This is demonstrated at electrophysiological level but also well shown using optogenetics in vivo and that this system can be “primed” and that activation can result in nocifensive behaviour outside of the stimulated dermatome. Overall an elegant array of techniques are used and I do think this data significantly advances our knowledge of nociceptive circuitry within the dorsal horn. I would like to see a number of technical areas clarified and also the conclusions are not yet fully supported by the data.

1) Methodological issues:

Was behavior performed with the observed blind to genotype? Generally this is good practice. Overall I think it is best that manuscripts such as this are reported in line with ARRIVE guidelines, which would avoid this kind of omission.

Materials and methods: “optogenetic probe surgery”. The way the results are presented (eg Figure 6C) the implication is that the optogenetic probe is implanted on one side of the spinal cord rather than midline. I may have missed it but I couldn't see this clearly stated in the Materials and methods. If so please state (and say which side). Are the characteristics of the probe such that you predict that you will only get activation of one side of the spinal cord? In implantation at level T12 and T13 I realize that the spinal cord below this vertebral level will be the lumbar enlargement. Can the authors state exactly which spinal segment that they are stimulating and whether they checked this (for instance the maximum level of fos activation see below).

2) The issue of crossing dermatomal boundaries and activation of a pro-excitatory network.

On the experiments where photostimulation is given at the level of T12-T13 and fos is assessed in anesthetized animals. The authors comment on the wide mediolateral extent of fos activation. What about rostro-caudal extent? It would be very interesting to know how many segments away from the primary stimulation site that fos activation is observed in the dorsal horn. This is clearly related to the behavioral findings presented in Figure 5 in which animals initially show a short latency nocifensive response to the foot this then extends to leg back and tail. Could the authors have looked at fos in these animals or was stimulation too short? Back to the point raised above on the siting of the optogenetic probe – in the hindlimbs was the response always ipsilateral to the side of stimulation? Please specify.

3) The contribution of CR neurons to sensitized pain states

Authors claim that CR neurons may contribute to allodynia. They could test this directly using photoinhibiton in these mice so that rather than just looking at baseline threshold change they investigate a situation in which allodynia is evoked (whether inflammatory or neuropathic). They also provide some evidence of temporal summation on repetitive stimulation and it would be helpful to consider a behavioral correlate of this and the response to photoinhibiton (eg. formalin test?).

4) Typos

Generally the paper is well written but I think they have some important statements the wrong way round and please check:

“In contrast photostimulation in a cohort of fiber optic probe implanted CReGFP (n=9) animals did produce photostimulation time locked behaviors…”

This is the control group surely you mean “did not”?

“in vivo photoinhibition significantly reduced paw withdrawal…”

I think you mean significantly increased?

Reviewer 3:

This study sets out to characterize the role of dorsal horn neurons that express Cre in calretinin-IRES-Cre (CR-cre) adult mice of both sexes using optogenetic tools. The authors used behavior analysis, electrophysiology and immunohistochemistry to show that CR-cre^+^ neurons relay information to lamina I projection neurons in the spinal cord and further in the brain to elicit pain. Using channelrhodopsin-2 assisted circuit mapping, they further show that CR-cre^+^ neurons form a highly interconnected network, some of them being capable of sustaining neuronal activity. The proposed manuscript is well written and findings are relevant for the neuroscience field, especially for research related to the somatosensory system and pain.

The main criticism that I would have for this otherwise excellent work is the use of CR-cre mice to drive the expression of tdTomato, ChR2 or Arch. Although I understand that the authors may not have had access to an inducible CR-creERT2 line, such as the one generated by Josh Huang, when they began this study, the data obtained here is sometimes a little difficult to interpret in regard to the CR^+^ population:

First, I disagree with the statement in the Discussion saying that the data was obtained from stimulated CR neurons, and not unidentified neurons that express CR transiently during development, based on co-expression of YFP and CR immunoreactivity. Indeed, for all experiments done with the CR-cre strain, these transient CR cells must also express the transgene and are necessary photostimulated/visible along with any other adult CR^+^ neurons. Actually, Figure 1 shows that 71% of YFP cells express CR, which might even be overestimated due to the difficulty of identifying YFP^+^ cell bodies in this particular mouseline, as exemplified in panel A.

Second, one cannot distinguish between light-induced responses generated by photoexcitation of spinal CR-cre^+^ interneurons from those induced by photoexcitation of central axon terminals coming from other CR^+^ cells (adult + transient) that are not spinal interneurons but which do project to the dorsal horn, such as primary sensory neurons. For example, some afferent fibers expressing CR innervate Pacinian corpuscles and send projections to the dorsal horn, including in the CR^+^ plexus in lamina II. Similarly, in the neuroanatomical section, tdTomato expression in CR-cre;Ai14 mice is not restricted to adult CR^+^ spinal interneurons but can also originate from transient CR^+^ neurons or other CR^+^ neurons that project to the dorsal horn.

Thus, unless the authors can reproduce some of the key results about CR neuronal connectivity and function in an inducible CR-creERT2 mouse in a timely fashion, I suggest a rewording thorough the whole manuscript that clearly considers this ambiguity, to avoid any misleading.

Subsection “CR^+^/SOM^+^ neurons provide direct input to Projection neurons”, authors conclude that most inputs onto NK1R^+^ cells originate from local interneurons based on the extensive colocalization between Homer immunoreactive puncta and VGLUT2 immunoreactivity. However, VGLUT2 is also expressed by numerous nociceptors, which likely synapse onto lamina I neurons. Similarly, somatostatin is expressed by spinal interneurons but also a subset of sensory neurons including pruriceptors which may synapse onto NK1R cells.

How do CR neurons sustain synaptic activity? While the answer of this question is beyond the scope of this study, this could have been addressed in the Discussion.

---

## [Author Response]

Reviewer 1:This is an interesting and incredibly detailed paper that addresses the contribution of calretinin neurons (presumably interneurons) in pain processing. The authors use a wealth of approaches and demonstrate quite convincingly that the great majority are excitatory, that they arborize extensively in superficial dorsal horn, that they engage many other CR neurons as well as non CR neurons, including some NK1 receptor expression neurons. The latter are presumed to be projection neurons. Based on electrophysiological and some behavioral analyses involving optogenetic activation of these neurons the authors conclude that the CR interneurons drive pain behavior by an excitatory action (including an interesting feed forward amplification that appears to prolong licking/biting) through these networks. The study adds considerably to many recent reports using somewhat comparable approaches that have studied the contribution of different interneuron populations to pain and itch processing.My concern reflects several issues, some of which might be addressable in a revision, but would require additional work.1) There are CR-expressing sensory neurons. For example, the Ernfors data base, finds expression of the gene in myelinated afferents (including proprioceptors). Although these afferents likely terminate in deep dorsal horn, it is always possible that there is some expression in afferents that terminate in superficial dorsal horn. That needs to be ruled out. At the minimum, tomato expression in sensory neurons should be reported.

While calretinin is known to be expressed in DRG neurons, transcriptomics studies suggest that expression is minimal and largely confined to the NF2 and NF 5 group myelinated afferents (Usoskin et al., 2015), and thus should not influence our experiments in the superficial dorsal horn. Regardless, we have directly assessed this issue, inspecting lumbar DRG neurons from CR^Cre^;Ai32 mice for YFP-expression. We find very few YFP cells in this tissue, but those we do find have large diameters. Importantly, these YFP cells all immunolabel for NF200, but do not express substance P. We have also used spinal cord sections from this mouse line to look at YFP expression in neurochemically defined populations of primary afferents in laminae I-V, and find virtually no overlap with VGLUT1-, VGLUT3-, SubP/CGRP-, or Pap-immunolabelled terminals. In addition, we have assessed the expression of calretinin (CR) immunolabelling in the central terminals of primary afferents labelled using an Advillin-GFP mouse line, and again find virtually no co-expression of CR and GFP in terminals from laminae I-IV. The only CR immunolabelling found in GFP-expressing afferent terminals was confined to deep medial lamina V, likely to account for the central arbors of CR-expressing DRG neurons. Nonetheless, these boutons were very sparse and rare but serve to reinforce that the pattern of central activation we see in our optogenetic experiments results from recruitment of spinal interneurons and not primary afferents. Taken together, these additional findings rule out a significant afferent contribution to the circuits we describe. We include this new data, and an accompanying figure, in our updated manuscript (subsection “Optogenetic activation of spinal CR^+^ neurons” paragraph three and Figure 1—figure supplement 2).

Finally, to exclude any role for peripheral afferent signalling in the in vivo poststimulation responses, we repeated photostimulation the responding paw was under local anaesthesia. Under these conditions not only did photostimulation-evoked responses remain intact, they were significantly longer in duration. This is consistent with the gate control theory that predicts the licking and grooming behaviours during photostimulation should diminish the optogenetic response. This information is now added to our manuscript (subsection “Photostimulation in behaving CR-ChR2 mice” paragraph two and Figure 6—figure supplement 1).

2) The cross that generated the pattern of expression was not tamoxifen-dependent. In other words, the pattern reflects embryonic Cre recombination. Can the authors be certain that they are dealing with the true CR pattern in the adult?

We undertook and reported due diligence assessments in the original manuscript for YFP overlap with calretinin in adult tissue (subsection “Optogenetic activation of spinal CR^+^ neurons” paragraph one and Figure 1—figure supplement 1). This demonstrated 70% of YFP cell profiles were calretinin positive. We agree that the remaining 30% of cells that express YFP, but not calretinin, may represent transient developmental calretinin expression. We now clearly explain this caveat in the manuscript and use the term calretinin-lineage when first introducing to the population we have studied (subsection “Optogenetic activation of spinal CR^+^ neurons” paragraphs one and two).

3) The authors only looked at mechanical stimulation. Is there any change in heat pain responsiveness? What about itch?

We elected to study mechanical stimulation as previous work, including our own, had highlighted this is the principle modality processed by calretinin neurons. This previous data also confirmed calretinin neurons are not implicated in circuits underlying noxious thermal processing (Seal, Goulding). In addition, the ability of morphine to abolish photostimulation responses argues against a role of these cells in itch, as morphine is not antipruritic. We now note these points in the Discussion and highlight the potential value of future studies to further address this topic (subsection “CR-ChR2 photostimulation responses are nociceptive and aversive” paragraph three and “Behavioral consequence of CR-ChR2 activation” paragraph one).

4) Most importantly, the authors only looked at acute pain behaviors provoked by activation of a large population of the CR neurons. It would be of great interest to understand their contribution, if any, in the setting of, for example, a model of neuropathic pain (partial sciatic nerve injury). Does DTR-mediated killing of cells, or inhibition of the cells optogenetically using the approaches described in the paper, alter mechanical hypersensitivity provoked in the setting of injury?

Our revised manuscript now reports data that speaks to this issue. Specifically, we present data from behavioural experiments that characterises the baseline response to photostimulation of the CR network, and then demonstrate that following spared nerve injury in these animals, photostimulation responses are potentiated. This neuropathic-related observation was not observed in sham operated animals. This new data is included in our updated manuscript (subsection “CR-ChR2 photostimulation responses are nociceptive and aversive” paragraph five and Figure 7 E-H).

5) Related to the latter question: What types of afferents engage the cells? Of particular interest are the Aβ afferents, which may carry the input that underlies mechanical allodynia in the setting of nerve injury.

We have not studied the afferent input to calretinin positive neurons directly in this manuscript but have previously reported that they respond to noxious mechanical stimulation, with some activation also resulting from hindpaw injection of capsaicin (Smith et al., 2015). Previous work has shown that A-β input can be relayed to calretinin cells by the slightly deeper positioned transient VGLUT3 population under pathological conditions (Peirs et al., 2015), as cited in our manuscript. This is consistent with a large literature showing that innocuous tactile input does not directly activate neurons in lamina II (where the calretinin population we report is located). We now make this information clearer in the manuscript (subsection “CR neuron microcircuits” paragraph two and “Behavioral consequence of CR-ChR2 activation” paragraph one).

Reviewer 2:

*The authors study the contribution of the Calretinin populations of interneurons within the dorsal horn using a Cre driver mouse line combining spinal electrophysiology, neuroanatomy combined optogenetics and behavioral assessment. There have been previous studies of this population of interneurons I think the advances are a more detailed assessment of the local circuitry of these interneurons and demonstration of a significant input onto lamina 1 projection neurons as well as the conclusion that although there is a small population of inhibitory CR interneurons the majority are excitatory and highly inter-connected acting in a feedforward manner to enhance excitatory signalling. This is demonstrated at electrophysiological level but also well shown using optogenetics* in vivo *and that this system can be “primed” and that activation can result in nocifensive behaviour outside of the stimulated dermatome. Overall an elegant array of techniques are used and I do think this data significantly advances our knowledge of nociceptive circuitry within the dorsal horn. I would like to see a number of technical areas clarified and also the conclusions are not yet fully supported by the data.*

1) Methodological issues:Was behavior performed with the observed blind to genotype? Generally this is good practice. Overall I think it is best that manuscripts such as this are reported in line with ARRIVE guidelines, which would avoid this kind of omission.

Yes, behaviour and of equal importance, subsequent video analysis of behaviour were both performed blind to genotype and condition throughout the study. Animal status was coded and then maintained confidential until the completion of experiments and analysis. This is now clearly stated in the manuscript (subsection “In vivo photostimulation and behaviour” paragraph one).

Materials and methods: “optogenetic probe surgery”. The way the results are presented (eg Figure 6C) the implication is that the optogenetic probe is implanted on one side of the spinal cord rather than midline. I may have missed it but I couldn't see this clearly stated in the Materials and methods. If so please state (and say which side). Are the characteristics of the probe such that you predict that you will only get activation of one side of the spinal cord? In implantation at level T12 and T13 I realize that the spinal cord below this vertebral level will be the lumbar enlargement. Can the authors state exactly which spinal segment that they are stimulating and whether they checked this (for instance the maximum level of fos activation see below).

Placement of probes was positioned lateral to the midline and over the T12-T13 intervertebral foramen. Our analysis of photostimulation-evoked cFos confirmed that activation was ipsilateral to probe placement and not contralateral. This was reported in Figure 3 and then the robust and reproducible behavioural response to photostimutation was used as evidence of probe placement in subsequent experiments. Probe placement is now further clarified in the manuscript (subsection “Optogenetic probe surgery”).

2) The issue of crossing dermatomal boundaries and activation of a pro-excitatory network.On the experiments where photostimulation is given at the level of T12-T13 and fos is assessed in anesthetized animals. The authors comment on the wide mediolateral extent of fos activation. What about rostro-caudal extent? It would be very interesting to know how many segments away from the primary stimulation site that fos activation is observed in the dorsal horn. This is clearly related to the behavioral findings presented in Figure 5 in which animals initially show a short latency nocifensive response to the foot this then extends to leg back and tail. Could the authors have looked at fos in these animals or was stimulation too short? Back to the point raised above on the siting of the optogenetic probe – in the hindlimbs was the response always ipsilateral to the side of stimulation? Please specify.

All animals were not prepared for cFos as this would require additional photostimulation, which was considered excessive once the initial fos experiment was complete. Other experimental objectives were then addressed. Regarding the rostrocaudal spread of fos labelling, this was not assessed. We note that mouse dermatomes are organised across the mediolateral as well as rostrocaudal planes, with distal limb regions represented in the medial cord and then more proximal limb and flank regions in the lateral cord. Much of the region relevant to nocifensive responses we report can be found in the L4/L5 spinal cord segments. Despite this, the behaviour response profile implies that recruitment of the calretinin network allows some degree of rostrocaudal spread of activation too, though not assessed in our data. This point is now raised in the Results and Discussion.

3) The contribution of CR neurons to sensitized pain statesAuthors claim that CR neurons may contribute to allodynia. They could test this directly using photoinhibiton in these mice so that rather than just looking at baseline threshold change they investigate a situation in which allodynia is evoked (whether inflammatory or neuropathic). They also provide some evidence of temporal summation on repetitive stimulation and it would be helpful to consider a behavioral correlate of this and the response to photoinhibiton (eg. formalin test?).

This relates to point 4 raised by reviewer one. As noted, our manuscript now includes an assessment of calretinin network activation under neuropathic pain conditions. This data shows that the behavioural response to spinal photostimulation is significantly enhanced following spared nerve injury (versus baseline response characterised prior to SNI). In contrast, photostimulation responses remain stable after the sham surgery. This new data is included in our updated manuscript (paragraph five subsection “CR-ChR2 photostimulation responses are nociceptive and aversive” and Figure 8 E-H).

4) TyposGenerally the paper is well written but I think they have some important statements the wrong way round and please check:“In contrast photostimulation in a cohort of fiber optic probe implanted CReGFP (n=9) animals did produce photostimulation time locked behaviors…”This is the control group surely you mean “did not”?

*“*in vivo *photoinhibition significantly reduced paw withdrawal…”*

I think you mean significantly increased?

Thank you for highlighting these editorial oversights, they are corrected in the revised manuscript.

Reviewer 3:This study sets out to characterize the role of dorsal horn neurons that express Cre in calretinin-IRES-Cre (CR-cre) adult mice of both sexes using optogenetic tools. The authors used behavior analysis, electrophysiology and immunohistochemistry to show that CR-cre^+^ neurons relay information to lamina I projection neurons in the spinal cord and further in the brain to elicit pain. Using channelrhodopsin-2 assisted circuit mapping, they further show that CR-cre^+^ neurons form a highly interconnected network, some of them being capable of sustaining neuronal activity. The proposed manuscript is well written and findings are relevant for the neuroscience field, especially for research related to the somatosensory system and pain.The main criticism that I would have for this otherwise excellent work is the use of CR-cre mice to drive the expression of tdTomato, ChR2 or Arch. Although I understand that the authors may not have had access to an inducible CR-creERT2 line, such as the one generated by Josh Huang, when they began this study, the data obtained here is sometimes a little difficult to interpret in regard to the CR^+^ population:First, I disagree with the statement in the Discussion saying that the data was obtained from stimulated CR neurons, and not unidentified neurons that express CR transiently during development, based on co-expression of YFP and CR immunoreactivity. Indeed, for all experiments done with the CR-cre strain, these transient CR cells must also express the transgene and are necessary photostimulated/visible along with any other adult CR^+^ neurons. Actually, Figure 1 shows that 71% of YFP cells express CR, which might even be overestimated due to the difficulty of identifying YFP^+^ cell bodies in this particular mouseline, as exemplified in panel A.Second, one cannot distinguish between light-induced responses generated by photoexcitation of spinal CR-cre^+^ interneurons from those induced by photoexcitation of central axon terminals coming from other CR^+^ cells (adult + transient) that are not spinal interneurons but which do project to the dorsal horn, such as primary sensory neurons. For example, some afferent fibers expressing CR innervate Pacinian corpuscles and send projections to the dorsal horn, including in the CR^+^ plexus in lamina II. Similarly, in the neuroanatomical section, tdTomato expression in CR-cre;Ai14 mice is not restricted to adult CR^+^ spinal interneurons but can also originate from transient CR^+^ neurons or other CR^+^ neurons that project to the dorsal horn.Thus, unless the authors can reproduce some of the key results about CR neuronal connectivity and function in an inducible CR-creERT2 mouse in a timely fashion, I suggest a rewording thorough the whole manuscript that clearly considers this ambiguity, to avoid any misleading.

The points raised here overlap with Reviewer 1 (points 1 and 2) and have been addressed in the manuscript. Briefly, our revised manuscript more clearly acknowledges the issue of transient CR expression during development and uses the term calretinin lineage, defined and subsequently abbreviated as CR-ChR2, to acknowledge this caveat. In addition, our revision includes an extensive analysis of CR inputs in the superficial dorsal horn, assessing any possible contribution from afferent populations that may also contribute to photostimulation responses. This data rules out a significant afferent contribution to the circuits we describe. CR lineage (paragraphs one and two subsection “Optogenetic activation of spinal CR+ neurons”). Afferent expression analysis (paragraph three and Figure 1—figure supplement 2).

Subsection “CR^+^/SOM^+^ neurons provide direct input to Projection neurons”, authors conclude that most inputs onto NK1R^+^ cells originate from local interneurons based on the extensive colocalization between Homer immunoreactive puncta and VGLUT2 immunoreactivity. However, VGLUT2 is also expressed by numerous nociceptors, which likely synapse onto lamina I neurons. Similarly, somatostatin is expressed by spinal interneurons but also a subset of sensory neurons including pruriceptors which may synapse onto NK1R cells.

We have elaborated on this aspect of the analysis by assessing the expression pattern of VGLUT2, SOM and CR in primary afferent terminals labelled in an Advillin::eGFP mouse line. We find virtually no evidence of either CR expression in GFP-labelled afferent terminals in laminae I-III. Likewise, Advillin::eGFP expression was largely absent from SOM and VGLUT2 immunolabelled profiles. Together, these additional data further support our original conclusion that synaptic inputs from CR, CR/SOM and SOM/VGLUT2 on to NK1R cells in lamina I are derived from interneurons. These labelling patterns are reported in our revised manuscript.

How do CR neurons sustain synaptic activity? While the answer of this question is beyond the scope of this study, this could have been addressed in the Discussion.

Our data suggests that the synaptically interconnected and excitatory nature of the CR network allows it to support continued (reverberatory) activity following strong stimulation such as that provided by optogenetic stimulation. These properties are compatible with our in vitro data showing evidence of short-term plasticity at these connections, as well as behavioural observations related to extended, enhanced, and spreading poststimulation responses in vivo. This explanation is now more clearly articulated in the Discussion.